

# Fluctuations of observables for free fermions in a harmonic trap at finite temperature

**Aurélien Grabsch⋆, Satya N. Majumdar, Grégory Schehr and Christophe Texier**

LPTMS, CNRS, Univ. Paris-Sud, Université Paris-Saclay, 91405 Orsay, France

⋆ aurelien.grabsch@u-psud.fr

## Abstract

We study a system of 1D non-interacting spinless fermions in a confining trap at finite temperature. We first derive a useful and general relation for the fluctuations of the occupation numbers valid for arbitrary confining trap, as well as for both canonical and grand canonical ensembles. Using this relation, we obtain compact expressions, in the case of the harmonic trap, for the variance of certain observables of the form of sums of a function of the fermions' positions, $\mathscr{L} = \sum_n h(x_n)$. Such observables are also called linear statistics of the positions. As anticipated, we demonstrate explicitly that these fluctuations do depend on the ensemble in the thermodynamic limit, as opposed to averaged quantities, which are ensemble independent. We have applied our general formalism to compute the fluctuations of the number of fermions $\mathscr{N}_+$ on the positive axis at finite temperature. Our analytical results are compared to numerical simulations. We discuss the universality of the results with respect to the nature of the confinement.


# 1 Introduction

The recent experimental progresses in cold atoms [1–3], which have made accessible new types of observables, have stimulated a renewed interest in the study of fermionic systems over the past few years. Although some emphasis has been put on many body physics, many physical aspects of the problem are captured by a simple non-interacting picture. Moreover, there are practical ways to reach the non-interacting regime experimentally [1,2,4]. Here, we will restrict ourselves to this case, and consider a system of $N$ fermions in a one dimensional trap described by the Hamiltonian

$$\hat{H} = \sum_{i=1}^{N} \left( \frac{\hat{p}_i^2}{2m} + V(\hat{x}_i) \right). \tag{1}$$

For specific choices of the confining potential $V(x)$, the positions of the fermions at zero temperature can be mapped onto the eigenvalues of random matrices. This relation is based on the fact that the ground state of this system takes the form of a Slater determinant:

$$\Psi_0(x_1, \ldots, x_N) = \frac{1}{\sqrt{N!}} \det[\psi_{i-1}(x_j)]_{1 \leqslant i,j \leqslant N}, \tag{2}$$

where $\psi_k(x)$ is the one-body eigenfunction of energy $\varepsilon_k$, with $k \in \mathbb{N}$. For example, in the case of a harmonic confinement $V(x) = \frac{1}{2}m\omega^2 x^2$, the joint distribution of the positions, given by

the modulus square of the many-body wave function, reads:

$$|\Psi_0(x_1,\dots,x_N)|^2 = \frac{1}{\mathscr{Z}_N} \prod_{i<j} |x_i - x_j|^2 \prod_{i=1}^{N} e^{-m\omega x_i^2/\hbar}, \tag{3}$$

where $\mathscr{Z}_N$ is a normalisation constant. It is exactly the distribution of the eigenvalues of matrices in the Gaussian unitary ensemble (GUE) [5,6]. Similarly, in the case of an infinite square well, the distribution of the positions can be mapped onto the Jacobi unitary ensemble (JUE) [7,8]. This connection to random matrices has allowed to study many properties of the ground state, like the density, the correlations, the number fluctuations and entanglement entropy [9–12]. It is interesting to remark that in the cold atom literature, some global results were derived using the local density approximation (LDA) [13–15], i.e. the Thomas-Fermi approximation, without realizing the connection to random matrix theory (RMT).

A remarkable recent achievement is the development of Fermi quantum microscopes [16–18], allowing the direct measurement of the fermions' positions in a confining trap. Motivated by this context, several theoretical studies have focused on different observables counting the fermions in a given spatial domain (see e.g. Refs. [9,10,12,19]). One such observable is the number $\mathscr{N}_+$ of fermions on the positive axis:

$$\mathscr{N}_+ = \sum_{n=1}^{N} \Theta(x_n), \quad \text{with} \quad \Theta(x) = \begin{cases} 1 & \text{if } x > 0, \\ 0 & \text{otherwise.} \end{cases} \tag{4}$$

Thanks to the mapping to RMT, the number $\mathscr{N}_+$ of fermions in the domain $x > 0$ at zero temperature is precisely the number of positive eigenvalues of GUE random matrices. In the RMT literature, this number is known as the *index*, and its statistical properties have been studied for various RMT ensembles, including the GUE [20,21] and the Cauchy ensemble [22]. For GUE, the mean value of $\mathscr{N}_+$ is trivially $N/2$, but the variance is nontrivial, given by

$$\text{Var}(\mathscr{N}_+)|_{T=0} \simeq \frac{1}{2\pi^2} \ln N + c, \qquad c = \frac{1+\gamma+3\ln 2}{2\pi^2}, \tag{5}$$

where $\gamma$ the Euler-Mascheroni constant. In the context of fermions, this finite value characterizes the quantum fluctuations. The logarithmic behaviour can be related to the anticorrelation of fermions discussed below, see Eq. (29). However in most experiments, the measurements are done at low but finite temperature. The zero temperature results obtained from RMT are therefore not sufficient to address these finite temperature properties. Our goal here will be to characterise the effect of thermal fluctuations on the variance of $\mathscr{N}_+$ and to generalise the result (5) to finite temperature.

Statistical physics provides several tools to analyse thermal fluctuations. When quantum correlations are dominant, like for the problem we aim to study, it is well known that the most efficient approach is supplied by the grand canonical ensemble in which the temperature $T$ and the chemical potential $\mu$ are fixed, while the energy and the number of fermions fluctuate. Many-body quantum eigenstates are conveniently labelled by a set of *occupation numbers* $\{n_k\}$, where $n_k = 1$ if the individual eigenstate $\psi_k(x)$ is occupied by one fermion and $n_k = 0$ otherwise. The grand canonical weight is:

$$\mathscr{P}_g(\{n_k\}) = \frac{1}{Z_g} e^{-\beta \sum_k n_k(\varepsilon_k - \mu)}, \tag{6}$$

where

$$Z_g(\varphi) = \sum_{\{n_k\}} \varphi^{\sum_k n_k} e^{-\beta \sum_k n_k \varepsilon_k} \tag{7}$$

is the grand canonical partition function. $\beta = 1/(k_B T)$ is the inverse temperature and $\varphi = e^{\beta\mu}$ the fugacity. In atomic traps, the number of atoms is fixed, which is best described by the microcanonical or canonical ensembles. Furthermore, due to the evaporative cooling techniques, the number of atoms is only moderately large, $N \sim 10^4$ to $10^7$, and the equivalence between statistical physics ensembles is questionable. This has recently motivated several works where basic quantities, such as occupation numbers, energy, specific heat, etc, have been analysed in the canonical and microcanonical ensembles (see for instance Refs. [23–28]). In the canonical ensemble, the number $N$ of fermions is fixed (instead of the chemical potential) and the Gibbs weight is

$$\mathscr{P}_c(\{n_k\}) = \frac{1}{Z_c} e^{-\beta \sum_k n_k \varepsilon_k} \delta_{\sum_k n_k, N}, \tag{8}$$

where

$$Z_c(N) = \sum_{\{n_k\}} e^{-\beta \sum_k n_k \varepsilon_k} \delta_{\sum_k n_k, N} \tag{9}$$

is the canonical partition function. The constraint on the number of particles included in the distribution however makes the calculations much more difficult in practice. For large $N$, deviations from the thermodynamic limit, and thus differences between predictions from the ensembles, are supposed to be small. However, it is worth stressing that the equivalence of ensembles is valid only for thermodynamic quantities (averaged observables), and not for their fluctuations [29, 30], which are our main interest here. In the present article, we will introduce a general formalism allowing to study the fluctuations of a wide class of observables of the form

$$\mathscr{L} = \sum_{n=1}^{N} h(x_n), \tag{10}$$

known as *linear statistics* of the positions of the fermions, where $h$ is any given function, not necessarily linear. For example, the potential energy in a harmonic trap corresponds to $h(x) = x^2$, whereas the index $\mathscr{N}_+$, under consideration here, corresponds to $h(x) = \Theta(x)$. Most theoretical studies of fluctuations at finite temperature were performed in the grand canonical ensemble [2, 11, 31–33], with the exception of Ref. [28], where the specific form $\sum_n x_n^2$ has allowed an exact calculation at all temperatures in both ensembles (see also Ref. [34]). Our aim here is to introduce a general framework to analyse the fluctuations of arbitrary linear statistics within both the grand canonical and canonical ensembles.

## 1.1 Summary of the main results

The fluctuations of the linear statistics $\mathscr{L} = \sum_n h(x_n)$ have two origins at finite temperature. For a given many-body quantum state $|\{n_k\}\rangle$, labelled by the occupation numbers, the positions $x_n$'s fluctuate due to quantum fluctuations. In addition, the occupation numbers $n_k$'s themselves fluctuate at finite temperature –this is the thermal fluctuations. To characterize these thermal fluctuations, we have found a general relation for the correlator of occupation numbers:

$$\overline{n_k n_l}^{c,g} = (\mp) \frac{e^{\beta \varepsilon_k} \overline{n_k}^{c,g} - e^{\beta \varepsilon_l} \overline{n_l}^{c,g}}{e^{\beta \varepsilon_k} - e^{\beta \varepsilon_l}}, \quad \begin{cases} - & \text{for bosons,} \\ + & \text{for fermions,} \end{cases} \tag{11}$$

where $\overline{\cdots}^{c,g}$ denotes the thermal average (see also [35]). We stress that this relation (11) is a very general one, independent of the confining trap. Moreover it is valid both in the canonical (c) and grand canonical ensemble (g). Note that in the grand canonical case, where the mean occupation numbers are given respectively by the Bose-Einstein and Fermi-Dirac distributions, this relation leads to: $\overline{n_k n_l}^g = \overline{n_k}^g \, \overline{n_l}^g$. This is the well-known independence of energy levels. This relation (11) plays a crucial role for the derivation of our subsequent results.

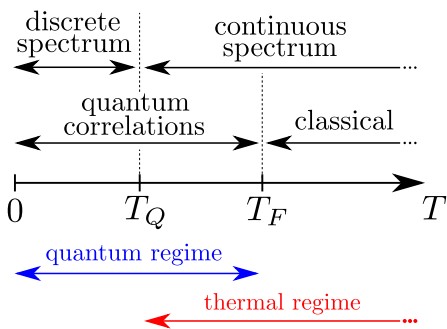

Figure 1: Sketch of the different temperature regimes.

We have studied a 1D system of $N$ fermions in a harmonic trap $V(x) = \frac{1}{2}m\omega^2 x^2$, where $N$ is either fixed (canonical ensemble) or fluctuating (grand canonical ensemble). We are interested in the fluctuations of the number $\mathcal{N}_+$ of particles in the domain $x > 0$. We can already identify two temperature scales, which will play a role below:

- a quantum scale $T_Q = \hbar\omega/k_B$: when $T \sim T_Q$, the small thermal energy allows only a few excitations near the Fermi level and the discreteness of the spectrum matters. We refer to this case as the *quantum regime*;

- the Fermi temperature $T_F = N\hbar\omega/k_B$: when $T \sim T_F$, the system is dominated by large thermal fluctuations. All energy levels contribute and the spectrum can be considered as continuous. Thus we call it the *thermal regime*.

The quantum regime covers the transition between the regime where the spectrum should be considered as discrete ($T \ll T_Q$) and the regime where it can be considered as continuous ($T \gg T_Q$), while the thermal regime describes the crossover between the regime dominated by quantum fluctuations ($T \ll T_F$) and the classical regime ($T \gg T_F$) [36], cf. Fig.1. Note that, while in the canonical ensemble $N$ is fixed, in the grand canonical ensemble it is a fluctuating random variable. Then, in the grand canonical ensemble, the Fermi temperature is defined as $T_F = \overline{N}^g \hbar\omega/k_B$. Henceforth, we will denote the variance of the total number of particles by $\mathrm{Var}_g(N)$ (the properties of this variance, in particular its $T$-dependence, are discussed in Appendix A; it is plotted in Fig. 4 as a function of temperature). We denote $\mathrm{Var}(\mathcal{N}_+)$ the variance of $\mathcal{N}_+$ which is computed in these two different regimes and the different statistical ensembles.

In the quantum regime $T \sim T_Q$, we have obtained:

$$\mathrm{Var}(\mathcal{N}_+) \simeq \begin{cases} \mathrm{Var}(\mathcal{N}_+)|_{T=0} + F_Q(\xi) & \text{canonical,} \\ \mathrm{Var}(\mathcal{N}_+)|_{T=0} + F_Q(\xi) + \dfrac{1}{4}\mathrm{Var}_g(N) & \text{grand canonical,} \end{cases} \tag{12}$$

where the zero temperature variance $\mathrm{Var}(\mathcal{N}_+)|_{T=0}$ is given by Eq. (5), and we introduced

$$F_Q\left(\xi = \frac{T_Q}{T}\right) = \frac{2}{\pi^2} \sum_{n=1}^{\infty} \frac{1}{2n-1} \frac{1}{e^{(2n-1)\xi} - 1}. \tag{13}$$

In the grand canonical case, the variance receives a contribution proportional to the fluctuations of the total number of particles, $\mathrm{Var}_g(N)$. This observation is also valid in the thermal regime $T \sim T_F$, where our result reads:

$$\mathrm{Var}(\mathcal{N}_+) \simeq \begin{cases} N F_T(y) & \text{canonical,} \\ \overline{N}^g F_T(y) + \dfrac{1}{4}\mathrm{Var}_g(N) & \text{grand canonical,} \end{cases} \tag{14}$$

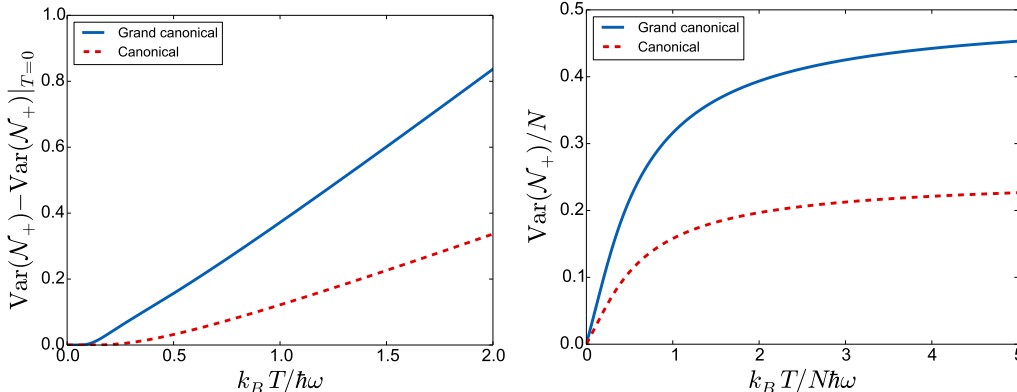

Figure 2: Variance of the number $\mathcal{N}_+$ of fermions in the domain $x > 0$ as a function of the temperature, both in the canonical and grand canonical ensembles. Left: quantum regime $T \sim T_Q$. Right: thermal regime $T \sim T_F$. In the grand canonical case, the number of particles $N$ must be replaced by its mean value $\overline{N}^g$.

where

$$F_T\left(y = \frac{T_F}{T}\right) = \frac{1 - e^{-y}}{4y} \, . \tag{15}$$

In the thermal regime, we have not included the contribution of the quantum fluctuations at zero temperature, as it is subdominant, of order $\ln N$. We checked that these two regimes smoothly match together for $T_Q \ll T \ll T_F$. Plots of these different expressions for the variance are shown in Fig. 2.

   We stress an important difference between the two scaling functions $F_Q$ and $F_T$: while the latter is not universal, i.e. depends on the precise form of the confining potential, the function $F_Q$ is universal.

## 1.2   Outline of the paper

The paper is organised as follows: Section 2 introduces the linear statistics for fermions. A general formula for the variance of linear statistics is obtained. In Section 3 we derive universal relations on the occupation numbers, which are useful to derive our results for the variance. We check our formulae in Section 4 by recovering the results of Ref. [28] on the potential energy. The case of the index variance for fermions in a harmonic well is discussed in Section 5. We end the paper with some concluding remarks. Some technical details are relegated to Appendices.

## 2   Observables of the form of linear statistics

Consider a system of spinless and non-interacting fermions in a confining potential $V(x)$. We denote $\psi_k(x)$ the one-particle wave-function associated to the energy $\varepsilon_k$, with $k \in \mathbb{N}$. These energy levels are non-degenerate in one dimension. In the absence of interaction, a many-body quantum state can be conveniently labelled by the set of occupation numbers $\{n_k\}$, where $n_k = 0$ or $1$ for fermions. The total number of fermions is then $N = \sum_k n_k$ fermions. The associated many-body wave function takes the form of a Slater determinant combining the one-particle wave functions of the occupied levels:

$$\Psi_{\{n_k\}}(x_1, \ldots, x_N) = \frac{1}{\sqrt{N!}} \det[\psi_{k_i}(x_j)]_{1 \leqslant i,j \leqslant N} \, , \tag{16}$$

where $\{k_i\}_{i=1,\ldots,N}$ is the list of occupied levels: $n_k = 1$ if $k \in \{k_i\}$.

We will consider the situation where the fermions are subjected to thermal fluctuations. In the following, we describe these fluctuations either in the *canonical* or the *grand canonical* ensemble. Since the description of our system of fermions involves two different sets of random variables, the positions and the occupation numbers, we are led to define two types of averaging:

- the "quantum average", denoted $\langle\cdots\rangle_{|\{n_k\}\rangle}$, which corresponds to averaging over the positions of the fermions. This averaging is defined for a given quantum state $|\{n_k\}\rangle$, with fixed number $N = \sum_k n_k$ of particles, as:

$$\langle F(x_1,\ldots,x_p)\rangle_{|\{n_k\}\rangle} = \int \mathrm{d}x_1 \ldots \mathrm{d}x_N \left|\Psi_{\{n_i\}}(x_1,\ldots,x_N)\right|^2 F(x_1,\ldots,x_p), \qquad (17)$$

for any function $F$ of $p \leqslant N$ positions. In the following we will often omit the subscript $|\{n_k\}\rangle$ for simplicity.

- the "thermal average", which corresponds to averaging over the occupation numbers. This averaging depends on the ensemble under consideration. We denote it $\overline{\cdots}^{\mathrm{c}}$ in the canonical ensemble and $\overline{\cdots}^{\mathrm{g}}$ in the grand canonical ensemble. For any function $G$ of the occupation numbers, it is defined as

$$\overline{G(\{n_k\})}^{\mathrm{c,g}} = \sum_{\{n_k\}} \mathscr{P}_{\mathrm{c,g}}(\{n_k\}) \, G(\{n_k\}), \qquad (18)$$

where $\mathscr{P}_{\mathrm{c,g}}(\{n_k\})$ represents the canonical or grand canonical measures given by Eqs. (8) and (6) respectively.

The quantum and thermal average will involve first averaging over the positions of the fermions $\{x_n\}$, Eq. (17), and then over the occupation numbers, Eq. (18). At zero temperature, the system is frozen in its ground state $\Psi_0$ and only the quantum average remains:

$$\overline{\langle F(x_1,\ldots,x_p)\rangle}^{\mathrm{c,g}}\bigg|_{T=0} = \int \mathrm{d}x_1 \ldots \mathrm{d}x_N \, |\Psi_0(x_1,\ldots,x_N)|^2 \, F(x_1,\ldots,x_p). \qquad (19)$$

Such integrals can be evaluated by making use of the determinantal structure of $\Psi_0$, as we will see in section 2.1. When going to finite temperature, the excited states contribute to the thermal averaging:

$$\overline{\langle F(x_1,\ldots,x_p)\rangle}^{\mathrm{c,g}} = \sum_{\{n_k\}} \mathscr{P}_{\mathrm{c,g}}(\{n_k\}) \int \mathrm{d}x_1 \ldots \mathrm{d}x_N \left|\Psi_{\{n_i\}}(x_1,\ldots,x_N)\right|^2 F(x_1,\ldots,x_p). \qquad (20)$$

The integral over the positions can still be computed by using the determinantal structure, but the summation over the quantum states makes the problem much more challenging.

In the following, we will study a wide class of observables which take the form of linear statistics $\mathscr{L}$ of positions of the fermions, Eq. (10). Our aim is to study the variance

$$\mathrm{Var}_{\mathrm{c,g}}(\mathscr{L}) = \overline{\langle\mathscr{L}^2\rangle}^{\mathrm{c,g}} - \left(\overline{\langle\mathscr{L}\rangle}^{\mathrm{c,g}}\right)^2. \qquad (21)$$

In order to compute this variance, we first need to evaluate the quantum averages $\langle\mathscr{L}\rangle$ and $\langle\mathscr{L}^2\rangle$ in a given quantum state $|\{n_k\}\rangle$. This is the object of the next section. The thermal averaging will be discussed in Section 3.

## 2.1 Quantum averages and determinantal structure

Let us first consider a quantum state $|\{n_k\}\rangle$, which contains $N = \sum_k n_k$ particles. It is convenient to use the fact that the positions of the fermions, in that given state, is a determinantal point process [11,33]. This means that the modulus square of the many-body wave function can be rewritten as a determinant

$$\left|\Psi_{\{n_k\}}(x_1,\dots,x_N)\right|^2 = \frac{1}{N!}\det\left[K(x_i,x_j;\{n_k\})\right]_{1\leqslant i,j\leqslant N}, \tag{22}$$

where we have introduced the kernel

$$K(x,y;\{n_k\}) = \sum_{k=0}^{\infty} n_k\,\psi_k^*(x)\psi_k(y). \tag{23}$$

Since the wave functions are orthogonal, this kernel verifies the reproducibility property

$$\int K(x,y;\{n_k\})K(y,z;\{n_k\})\mathrm{d}y = K(x,z;\{n_k\}), \tag{24}$$

where we used that $n_k^2 = n_k$ since $n_k = 0$ or $1$ for fermions. The direct consequence of this property is that the $n$-points correlation function also takes a simple determinantal form:

$$R_n(x_1,\dots,x_n) = \frac{N!}{(N-n)!}\int \mathrm{d}x_{n+1}\dots\mathrm{d}x_N\left|\Psi_{\{n_k\}}(x_1,\dots,x_N)\right|^2$$

$$= \det\left[K(x_i,x_j;\{n_k\})\right]_{1\leqslant i,j\leqslant n}. \tag{25}$$

In particular, the one-point function is given by

$$R_1(x) = N\int \mathrm{d}x_2\dots\mathrm{d}x_N\left|\Psi_{\{n_k\}}(x_1,\dots,x_N)\right|^2 = K(x,x;\{n_k\}). \tag{26}$$

When averaged over $n_k$'s in the grand canonical ensemble, using Eq. (6), this is precisely the mean density of fermions

$$\overline{\langle\rho(x)\rangle}^{\mathrm{g}} = K(x,x;\{\overline{n_k}^{\mathrm{g}}\}) \tag{27}$$

where $\rho(x) = \sum_n \delta(x-x_n)$ and $\overline{\langle\cdots\rangle}^{\mathrm{g}}$ is the usual quantum and statistical average. The two-point correlation function reads:

$$R_2(x,y) = K(x,x;\{n_k\})K(y,y;\{n_k\}) - K(x,y;\{n_k\})^2. \tag{28}$$

When averaged over $n_k$'s in the grand canonical ensemble, this is related to the familiar relation for the density-density correlation function in the Fermi gas

$$\overline{\langle\rho(x)\rho(y)\rangle}^{\mathrm{g}}_{\mathrm{corr}} = \delta(x-y)\overline{\langle\rho(x)\rangle}^{\mathrm{g}} - \left|K(x,y;\{\overline{n_k}^{\mathrm{g}}\})\right|^2, \tag{29}$$

where the minus sign is related to the effective repulsion between fermions due to the Pauli principle. When the density can be considered constant, equal to $\bar\rho$, the zero temperature kernel is the famous sine-kernel $K(x,y)|_{T=0} = \bar\rho\,\mathrm{sinc}[\bar\rho\pi(x-y)]$, where $\mathrm{sinc}(z) = \sin z/z$, related to the anti-correlations

$$\langle\rho(x)\rho(y)\rangle_{\mathrm{corr}} = \bar\rho\,\delta(x-y) - \bar\rho^2\,\mathrm{sinc}^2[\bar\rho\pi(x-y)] \qquad \text{(in bulk)}. \tag{30}$$

Using these properties we can easily express the quantum average of the linear statistics (10):

$$\langle \mathscr{L} \rangle_{|\{n_k\}\rangle} = \left\langle \sum_{n=1}^{N} h(x_n) \right\rangle = N \langle h(x_1) \rangle \,, \tag{31}$$

where we used the symmetry under the exchange of particles. Therefore, only the one-point function, Eq. (26), is needed for this computation:

$$\langle \mathscr{L} \rangle_{|\{n_k\}\rangle} = \int h(x) R_1(x) \mathrm{d}x = \int h(x) K(x, x, \{n_k\}) \mathrm{d}x \,. \tag{32}$$

Using the expression of the kernel, Eq. (23), one can express this average in terms of the one-particle wave functions:

$$\langle \mathscr{L} \rangle_{|\{n_k\}\rangle} = \sum_k n_k \int h(x) |\psi_k(x)|^2 \, \mathrm{d}x \,. \tag{33}$$

We can similarly compute the mean square:

$$\begin{aligned}
\langle \mathscr{L}^2 \rangle_{|\{n_k\}\rangle} &= N \langle h(x_1)^2 \rangle + N(N-1) \langle h(x_1) h(x_2) \rangle \\
&= \int h(x)^2 R_1(x) \, \mathrm{d}x + \int h(x) h(y) R_2(x, y) \, \mathrm{d}x \mathrm{d}y \\
&= \sum_k n_k B_k + \sum_{k,l} n_k n_l (A_{k,k} A_{l,l} - (A_{k,l})^2) \,,
\end{aligned} \tag{34}$$

where we have introduced the matrix elements

$$A_{k,l} = \int h(x) \psi_k^*(x) \psi_l(x) \mathrm{d}x \,, \quad B_k = \int h(x)^2 |\psi_k(x)|^2 \, \mathrm{d}x \,. \tag{35}$$

Using these definitions, we can rewrite (33) as:

$$\langle \mathscr{L} \rangle_{|\{n_k\}\rangle} = \sum_k n_k A_{k,k} \,. \tag{36}$$

This derivation shows that one only needs to compute the matrix elements $A_{k,l}$ and $B_k$ to perform the quantum averages involved in the variance of a linear statistics $\mathscr{L}$.

## 2.2 A general formula for the variance

Using the results of the quantum average, Eqs. (34,36), one can straightforwardly take the thermal average

$$\overline{\langle \mathscr{L} \rangle}^{c,g} = \sum_k \overline{n_k}^{c,g} A_{k,k} \,, \tag{37}$$

$$\overline{\langle \mathscr{L}^2 \rangle}^{c,g} = \sum_k \overline{n_k}^{c,g} B_k + \sum_{k,l} \overline{n_k n_l}^{c,g} (A_{k,k} A_{l,l} - (A_{k,l})^2) \,, \tag{38}$$

in the canonical or grand canonical ensemble. Combining these two relations, we obtain a general expression for the variance of a linear statistics:

$$\boxed{\mathrm{Var}_{c,g}(\mathscr{L}) = \sum_k \overline{n_k}^{c,g} B_k - \sum_{k,l} \overline{n_k}^{c,g} \overline{n_l}^{c,g} (A_{k,l})^2 + \sum_{k \neq l} \mathrm{Cov}_{c,g}(n_k, n_l)(A_{k,k} A_{l,l} - (A_{k,l})^2)} \tag{39}$$

where $\text{Cov}_{c,g}(n_k, n_l) = \overline{n_k n_l}^{c,g} - \overline{n_k}^{c,g}\overline{n_l}^{c,g}$. The difference between the two ensembles clearly appears on this relation. Indeed, the covariance of occupation numbers is zero in the grand canonical ensemble: $\text{Cov}_g(n_k, n_l) = 0$ for $k \neq l$. We will see that this term gives an additional contribution of the same order as the first two.

In order to use this formula for the variance, one first needs to compute the coefficients $A_{k,l}$ and $B_k$, which will be analysed in Appendix C. Then, the sums over the levels need to be evaluated. An important piece of the calculation is a relation on the occupation numbers which will be derived in the next section.

# 3   Occupation numbers

In this section, we derive general relations involving the occupation numbers, valid both in the canonical and grand canonical ensembles. Although this article focuses on fermionic systems, we will also consider the bosonic case for the sake of generality.

## 3.1   Grand canonical ensemble

Let us first consider the case of the grand canonical ensemble. In this ensemble, the mean occupation numbers are the well-known Bose-Einstein and Fermi-Dirac distributions:

$$\overline{n_k}^g = \begin{cases} \dfrac{1}{e^{\beta(\varepsilon_k - \mu)} - 1} & \text{for bosons, with } \mu < \varepsilon_0 \\[2ex] \dfrac{1}{e^{\beta(\varepsilon_k - \mu)} + 1} & \text{for fermions} \end{cases} \tag{40}$$

where $\mu$ is the chemical potential, which controls the mean number of particles

$$\overline{N}^g = \sum_k \overline{n_k}^g \,. \tag{41}$$

The variance of the occupation numbers is also known:

$$\text{Var}_g(n_k) = \overline{(n_k^2)}^g - (\overline{n_k}^g)^2 = \overline{n_k}^g(1 \pm \overline{n_k}^g)\,, \tag{42}$$

with the upper sign for bosons and the lower sign for fermions. The simplicity and the success of the grand canonical ensemble relies on the independence of individual energy levels, i.e. the absence of correlations between occupation numbers:

$$\text{Cov}_g(n_k, n_l) = \overline{n_k n_l}^g - \overline{n_k}^g\, \overline{n_l}^g = 0 \quad \text{if} \quad k \neq l\,. \tag{43}$$

## 3.2   Canonical ensemble

In the canonical ensemble, the total number of particles is fixed to $N$. This constraint induces correlations between the occupation numbers associated to different levels, which are in general very difficult to handle. The mean canonical occupation numbers are obtained by averaging with the canonical measure (8):

$$\overline{n_k}^c = \sum_{\{n_i\}} n_k\, \mathscr{P}_c(\{n_i\}) = \frac{1}{Z_c(N)} \sum_{\{n_i\}} n_k\, e^{-\beta \sum_i n_i \varepsilon_i}\, \delta_{\sum_i n_i, N} \,. \tag{44}$$

The general strategy of statistical physics is to introduce generating functions, i.e. consider the sum

$$\sum_{N=0}^{\infty} \varphi^N \sum_{\{n_i\}} n_k\, e^{-\beta \sum_i n_i \varepsilon_i}\, \delta_{\sum_i n_i, N} = Z_g(\varphi) \sum_{\{n_i\}} n_k\, \mathscr{P}_g(\{n_i\}) = Z_g(\varphi)\, \overline{n_k}^g \,, \tag{45}$$

where we recognised the grand canonical measure, Eq. (6). We can then deduce $\overline{n_k}^c$ by a contour integral which selects the $\varphi^N$ term in Eq. (45):

$$\overline{n_k}^c = \frac{1}{Z_c(N)} \oint \frac{d\varphi}{2i\pi} \frac{Z_g(\varphi)}{\varphi^{N+1}} \overline{n_k}^g \, , \tag{46}$$

where the integrals run over a closed contour winding once around the origin in the counter-clockwise direction. The partition functions $Z_c$ and $Z_g$ being related by

$$Z_g(\varphi) = \sum_N \varphi^N Z_c(N) \, , \qquad Z_c(N) = \oint \frac{d\varphi}{2i\pi} \frac{Z_g(\varphi)}{\varphi^{N+1}} \, , \tag{47}$$

we can rewrite the occupation numbers as

$$\overline{n_k}^c = \frac{\oint \dfrac{d\varphi}{2i\pi} \dfrac{Z_g(\varphi)}{\varphi^{N+1}} \overline{n_k}^g}{\oint \dfrac{d\varphi}{2i\pi} \dfrac{Z_g(\varphi)}{\varphi^{N+1}}} \, . \tag{48}$$

A similar relation clearly holds for any average quantity, for instance

$$\overline{n_k n_l}^c = \frac{1}{Z_c(N)} \oint \frac{d\varphi}{2i\pi} \frac{Z_g(\varphi)}{\varphi^{N+1}} \overline{n_k n_l}^g = \frac{\oint \dfrac{d\varphi}{2i\pi} \dfrac{Z_g(\varphi)}{\varphi^{N+1}} \overline{n_k n_l}^g}{\oint \dfrac{d\varphi}{2i\pi} \dfrac{Z_g(\varphi)}{\varphi^{N+1}}} \, . \tag{49}$$

We will first derive general expressions for (48,49). In a second step we will analyse the large $N$ limit by a saddle point method. The two results will be useful in the next sections.

### 3.2.1 General representation for the covariance

To evaluate the numerator in Eq. (48), we only need to determine the coefficient of the term $\varphi^N$ in the expansion of $Z_g(\varphi)\overline{n_k}^g$ as a power series in $\varphi$. This series can be obtained by using Eqs. (40,47). Then, isolating the term proportional to $\varphi^N$ yields:

$$\overline{n_k}^c = \sum_{p=1}^{N} (\pm 1)^{p-1} \frac{Z_c(N-p)}{Z_c(N)} e^{-p\beta\varepsilon_k} \, , \tag{50}$$

where the upper sign is for bosons, and the lower sign for fermions. This relation was known in the literature, see e.g. Refs. [26, 37, 38]. Similarly, we can obtain the expression for the product of occupation numbers:

$$\overline{n_k n_l}^c = \sum_{p=1}^{N-1} \sum_{q=1}^{N-p-1} (\pm 1)^{p+q} \frac{Z_c(N-p-q)}{Z_c(N)} e^{-p\beta\varepsilon_k} e^{-q\beta\varepsilon_l} \, , \quad \text{for} \quad k \neq l \, . \tag{51}$$

This last relation can be found in the case of bosons in Ref. [26]. It can be simplified by introducing $s = p + q$:

$$\overline{n_k n_l}^c = \sum_{s=2}^{N-1} (\pm 1)^s \frac{Z_c(N-s)}{Z_c(N)} e^{-\beta s \varepsilon_l} \sum_{p=1}^{s-1} e^{-\beta p (\varepsilon_k - \varepsilon_l)}$$

$$= \sum_{s=2}^{N} (\pm 1)^s \frac{Z_c(N-s)}{Z_c(N)} \frac{e^{-\beta(s-1)\varepsilon_l} - e^{-\beta(s-1)\varepsilon_k}}{e^{\beta\varepsilon_k} - e^{\beta\varepsilon_l}} \, . \tag{52}$$

Separating this expression into two sums, we recognise the expressions of $\overline{n_k}^c$ and $\overline{n_l}^c$, Eq. (50), thus,

$$\overline{n_k n_l}^c = (\mp) \frac{e^{\beta \varepsilon_k} \overline{n_k}^c - e^{\beta \varepsilon_l} \overline{n_l}^c}{e^{\beta \varepsilon_k} - e^{\beta \varepsilon_l}} . \tag{53}$$

We derived this relation in the canonical ensemble, but one can easily check that it also holds in the grand-canonical one, where it becomes $\overline{n_k n_l}^g = \overline{n_k}^g \overline{n_l}^g$. Therefore, this is a very general relation, valid for any system of non interacting bosons or fermions, either in the canonical or grand canonical ensemble.

We have extended this analysis to the $p$-point correlation functions in Ref. [35].

### 3.2.2  Saddle point estimate for large $N$

Despite having obtained general relations (50,51), their large $N$ analysis remains a challenge. For this purpose, we perform a saddle point analysis of Eqs. (48,49). For all the integrals, the saddle point $\varphi_\star$ is given by the condition

$$\sum_k \overline{n_k}^g = N , \tag{54}$$

which fixes the fugacity (or the chemical potential) such that the grand canonical mean number of particles in the trap is equal to the fixed canonical number $N$. Then, the integral (48) yields:

$$\overline{n_k}^c = \overline{n_k}^g + \mathcal{O}(N^{-1}) . \tag{55}$$

This indicates that the statistics of the occupation numbers are the same at leading order in $N$ in the canonical and grand-canonical ensembles. This is not surprising since we expect both ensembles to be equivalent for averaged quantities in the thermodynamic limit. The equivalence of thermodynamic results however only holds for averages and not for variances or covariances, as it is well-known [29]. Performing the same saddle point analysis with Eq. (49) yields

$$\overline{n_k n_l}^c = \overline{n_k}^g \, \overline{n_l}^g + \mathcal{O}(N^{-1}) . \tag{56}$$

Since we will need the covariance of occupation numbers, we push the saddle point approximation of Eqs. (48,49) to the next order, see Appendix B, in order to get the $\mathcal{O}(N^{-1})$ corrections. After many simplifications, we obtain the compact expression:

$$\text{Cov}_c(n_k, n_l) = -\frac{\text{Var}_g(n_k)\text{Var}_g(n_l)}{\sum_p \text{Var}_g(n_p)} + \mathcal{O}(N^{-2}) , \tag{57}$$

where $\text{Var}_g(n_k)$ is given by Eq. (42). Note that since $\text{Var}_g(n_k) = \text{Var}_c(n_k) + \mathcal{O}(N^{-1})$, we can express this covariance in terms of the variance of occupation numbers in any ensemble. We chose to express it in terms of the grand canonical one since the expressions are simpler in this case. In the canonical ensemble, $N = \sum_k n_k$ is fixed, which implies the sum rule

$$\sum_k \text{Var}_c(n_k) + \sum_{k \neq l} \text{Cov}_c(n_k, n_l) = 0 . \tag{58}$$

It is clear that Eq. (57) is consistent with this sum rule, up to $\mathcal{O}(N^0)$.

However relation (57) is valid as long as the covariance is a $\mathcal{O}(N^{-1})$ correction to the saddle point approximation. It is the case when the denominator, which corresponds to the variance of the total number of particles in the grand-canonical ensemble, is of order $N$: $\sum_p \text{Var}_g(n_p) = \text{Var}_g(N) = \mathcal{O}(N)$. As discussed in Appendix A, this is verified only in the thermal regime $T \sim T_F$. Therefore, this relation for the covariance can only be used this regime. In the quantum regime $T \sim T_Q$ we will instead rely on the exact relation (53).

### 3.3 A symmetry relation for fermion occupation numbers

Let us now consider fermions in a harmonic trap $V(x) = \frac{1}{2}m\omega^2 x^2$, which will be the main focus of the paper. In this case, the spectrum is linear, $\varepsilon_n = (n + \frac{1}{2})\hbar\omega$, $n \in \mathbb{N}$.

In the grand canonical ensemble, the Fermi-Dirac distribution (40) has a the well-known symmetry around the chemical potential $\mu$:

$$\frac{1}{e^{\beta(\varepsilon - \mu)} + 1} = 1 - \frac{1}{e^{-\beta(\varepsilon - \mu)} + 1} , \tag{59}$$

which is usually interpreted as the particle-hole symmetry. In the case of a discrete spectrum, setting the chemical potential in the middle of a gap, $\mu = N_f \hbar\omega$, where $N_f$ is an integer, the relation reduces to:

$$\overline{n_{N_f + k}}^{\,\mathrm{g}} = 1 - \overline{n_{N_f - k - 1}}^{\,\mathrm{g}} . \tag{60}$$

Due to the linearity of the spectrum, we have $\overline{N}^{\,\mathrm{g}} = N_f$ up to exponentially small correction $\sim (k_B T / \hbar\omega) e^{-N_f \hbar\omega/(k_B T)}$.

In the canonical ensemble the occupation numbers (50), are expressed in terms of the canonical partition function. For the harmonic oscillator, it can be computed analytically, and the result can be found in a few textbooks [30, 39]:

$$Z_c(N) = e^{-\frac{N^2 \beta\hbar\omega}{2}} \prod_{n=1}^{N} \frac{1}{1 - e^{-n\beta\hbar\omega}} . \tag{61}$$

Using this result, the expression of the occupation numbers (50) greatly simplifies near the Fermi level in the large $N$ limit:

$$\overline{n_{N+k}}^{\,\mathrm{c}} \simeq \sum_{p=1}^{\infty} (-1)^{p-1} e^{-\frac{\beta\hbar\omega}{2} p(2k + p + 1)} , \tag{62}$$

with corrections exponentially small with $N$. Using this expression, it is straightforward to show that the canonical occupation numbers also exhibit the particle-hole symmetry around the Fermi level:

$$\overline{n_{N+k}}^{\,\mathrm{c}} = 1 - \overline{n_{N-k-1}}^{\,\mathrm{c}} , \tag{63}$$

which is exactly the same as the grand canonical relation (60).

These symmetries (60,63), along with relations (53,57) on the occupation numbers will be essential for our study of linear statistics for fermions in a harmonic trap. We will first check our approach by recovering the recent results of Ref. [28] on the potential (or kinetic) energy of the fermions. Then, we will apply our method to the observable introduced in Section 1: the number $\mathcal{N}_+$ of fermions on the positive axis, given by Eq. (4).

## 4 A first check: potential energy of fermions in a harmonic trap

In the harmonic trap, the one-particle wave functions are expressed in terms of Hermite polynomials:

$$\psi_n(x) = \sqrt{\frac{\alpha}{2^n n! \sqrt{\pi}}} \, H_n(\alpha x) \, e^{-\alpha^2 x^2/2} , \quad \alpha = \sqrt{\frac{m\omega}{\hbar}} , \tag{64}$$

with energies $\varepsilon_n = (n + 1/2)\hbar\omega$, $n \in \mathbb{N}$. We denote $\{x_n\}$ the positions of the fermions.

Recently, the distribution of the potential energy $E_p$, or equivalently the kinetic energy, was obtained in Ref. [28] for any temperature $T$ or fixed number $N$ of fermions (canonical ensemble). However their method is restricted to the study of the potential energy

$$E_p = \frac{1}{2}m\omega^2 I, \qquad I = \sum_{n=1}^{N} x_n^2, \tag{65}$$

which is a specific linear statistics (10), with $h(x) = x^2$. They have obtained two different scaling functions describing the quantum and thermal regimes:

$$\text{Var}_c(I) \simeq \frac{N^2}{2\alpha^4} V_q\left(\frac{T}{T_Q}\right) \qquad \text{for } T \sim T_Q \tag{66}$$

and

$$\text{Var}_c(I) \simeq \frac{N^3}{2\alpha^4} V_{\text{th}}\left(\frac{T}{T_F}\right) \qquad \text{for } T \sim T_F \tag{67}$$

where the two scaling functions are

$$V_q(z) = \coth\frac{1}{z}, \tag{68}$$

$$V_{\text{th}}(z) = z\left[-6z^2 \text{Li}_2(1 - e^{1/z}) - 1 - \coth\frac{1}{2z}\right], \tag{69}$$

where $\text{Li}_s(z) = \sum_{k=0}^{\infty} z^k/k^s$ is the polylogarithm function. One can check that these two expressions (66,67) smoothly match in the intermediate regime $T_Q \ll T \ll T_F = N T_Q$, as the two scaling function present the limiting behaviours $V_q(z) \simeq z$ for $z \to \infty$ and $V_{\text{th}}(z) \simeq z$ for $z \to 0$. In this section we will check using our more general approach that we recover these results.

In this particular case of linear statistics with $h(x) = x^2$, the matrix elements (35) can be computed exactly:

$$B_k = \frac{3}{2\alpha^4}\left(k^2 + k + \frac{1}{2}\right), \tag{70}$$

$$A_{k,l} = \frac{1}{\alpha^2}\left\{\left(k + \frac{1}{2}\right)\delta_{k,l} + \frac{1}{2}\sqrt{(k+1)(k+2)}\,\delta_{k+2,l} + \frac{1}{2}\sqrt{k(k-1)}\,\delta_{k-2,l}\right\}. \tag{71}$$

## 4.1 Quantum regime

In this regime, the small temperature $T \sim T_Q$ allows only a few excitations above the Fermi level $\varepsilon_N$. Therefore, we expect the main contribution to come from the proximity of this level. At leading order in $N$, for fixed $k$ and $l$, the matrix elements become:

$$B_{N+k} \simeq \frac{3N^2}{2\alpha^4}, \tag{72}$$

$$A_{N+k,N+l} \simeq \frac{N\hbar}{m\omega}\left\{\delta_{k,l} + \frac{1}{2}\delta_{k+2,l} + \frac{1}{2}\delta_{k-2,l}\right\}. \tag{73}$$

Using these expressions in Eq. (39), the variance of $I$ becomes, at leading order:

$$\text{Var}_c(I) \simeq \frac{N^2}{4\alpha^4} \sum_{k=-\infty}^{\infty} \left(2\,\overline{n_{N+k}}^c - \overline{n_{N+k}n_{N+k-2}}^c - \overline{n_{N+k}n_{N+k+2}}^c\right), \tag{74}$$

where we extended the summation to $-\infty$ instead of $-N$, since the corrections are exponentially small. Using then relation (53), this becomes

$$\text{Var}_c(I) \simeq \frac{N^2}{4\alpha^4} \coth(\beta\hbar\omega) \sum_{k=-\infty}^{+\infty} \left(\overline{n_{N+k}}^c - \overline{n_{N+k+2}}^c\right). \tag{75}$$

This last sum cannot be separated into two sums because they would both diverge. Using the symmetry of the mean occupation numbers around the Fermi level, see Eq. (63), we get

$$\sum_{k=-\infty}^{\infty} \left(\overline{n_{N+k}}^c - \overline{n_{N+k+2}}^c\right) = \overline{n_{N-2}}^c - \overline{n_N}^c + \overline{n_{N-1}}^c - \overline{n_{N+1}}^c + 2\sum_{k=0}^{\infty}\left(\overline{n_{N+k}}^c - \overline{n_{N+k+2}}^c\right). \tag{76}$$

Under this form, the sum can be separated into two sums. This yields

$$\sum_{k=-\infty}^{\infty} \left(\overline{n_{N+k}}^c - \overline{n_{N+k+2}}^c\right) = 2(\overline{n_N}^c + \overline{n_{N+1}}^c) + \overline{n_{N-2}}^c - \overline{n_N}^c + \overline{n_{N-1}}^c - \overline{n_{N+1}}^c. \tag{77}$$

Finally, using again Eq. (63) gives

$$\sum_{k=-\infty}^{\infty} \left(\overline{n_{N+k}}^c - \overline{n_{N+k+2}}^c\right) = 2. \tag{78}$$

Therefore, we recover the result of Ref. [28] in the quantum regime, Eq. (66):

$$\text{Var}_c(I) \simeq \frac{N^2}{2\alpha^4} \coth(\beta\hbar\omega). \tag{79}$$

## 4.2 Thermal regime

We now consider the regime where the temperature is of the order of the Fermi temperature $T \sim T_F$. We fix $y = \beta N\hbar\omega = T_F/T$ and let $N \to \infty$. In this case, we can use the results of section 3.2.2, for instance

$$\overline{n_k}^c \simeq \overline{n_k}^g + \mathcal{O}(N^{-1}) = \frac{1}{e^{\beta(\varepsilon_k-\mu)}+1} + \mathcal{O}(N^{-1}), \tag{80}$$

where the chemical potential $\mu$ is fixed by $\sum_k \overline{n_k}^g = N$. This sum over $k$ can be replaced by an integral over $x = k/N$ since the discreteness of the spectrum plays no role in this regime:

$$\sum_k \overline{n_k}^g \simeq N \int_0^\infty \frac{dx}{e^{y(x-\mu/(N\hbar\omega))}+1} = \frac{N}{y} \ln\left(1 + e^{y\mu/N\hbar\omega}\right). \tag{81}$$

Imposing that this sum is the total number of particles yields:[1]

$$\mu = N\hbar\omega + \frac{N\hbar\omega}{y} \ln(1-e^{-y}) + \mathcal{O}(N^0), \quad y = \frac{T_F}{T}. \tag{82}$$

Therefore, we can rewrite the occupation numbers (80) as

$$\overline{n_k}^c \simeq f_y(k/N) + \mathcal{O}(N^{-1}), \tag{83}$$

---

[1] A more precise treatment of the steepest descent equation $\sum_k \overline{n_k}^g = N$, with the help of the Euler-MacLaurin formula $\sum_{k=0}^{\infty} f(k) = \int_0^\infty dx\, f(x) - \sum_{p=1}^{\infty} \frac{B_p}{p!} f^{(p-1)}(0)$ with $B_p$ the Bernoulli number ($B_1 = -1/2$, $B_2 = 1/6$, etc), shows that the zero temperature limit of the chemical potential is precisely in the middle of the energy gap above the last occupied level: $\mu = N\hbar\omega + \mathcal{O}(N^{-1}) = (\varepsilon_N + \varepsilon_{N-1})/2$.

where we introduced the notation

$$f_y(x) = \frac{1}{\frac{e^{y(x-1)}}{1-e^{-y}} + 1} \tag{84}$$

for the Fermi-Dirac distribution in terms of convenient variables. In most cases, the approximation (83) is sufficient. However, the corrections are essential when the fluctuations of the occupation numbers contribute. For instance, the covariances in the last term of Eq. (39) must be evaluated using Eq. (57). Replacing now the sums over $k$ by integrals over $x = k/N$, and keeping only the leading order of the matrix elements (70,71) yields:

$$\text{Var}_c(I) \simeq \frac{N^3}{\alpha^4} \int_0^\infty \left( \frac{3}{2} f_y(x) - f_y(x)^2 \right) x^2 \, dx - \frac{N}{2\alpha^4} \int_0^\infty f_y(x)^2 x^2 \, dx$$
$$- \frac{N^3}{\alpha^4} \frac{\left( \int_0^\infty f_y(x)(1-f_y(x)) x \, dx \right)^2}{\int_0^\infty f_y(x)(1-f_y(x)) \, dx}. \tag{85}$$

Evaluating these integrals, we finally obtain

$$\text{Var}_c(I) \simeq \frac{N^3}{2\alpha^4 y} \left( -\frac{6}{y^2} \text{Li}_2(1-e^y) - 1 - \coth \frac{y}{2} \right) \quad \text{for } y = \frac{T_F}{T}. \tag{86}$$

Hence we recover the result of Ref. [28] in the thermal regime, Eq. (67). This verification validates our approach to compute the variance of linear statistics.

### 4.3 Discussion

It is interesting to comment on the physical content of these results, and in particular compare the fluctuations of the potential energy with the fluctuations of the total energy $E = E_c + E_p$, which, in the canonical ensemble, can be related to the heat capacity studied in detail in [30] by $\text{Var}_c(E) = k_B T^2 C_V$. From (61), we get

$$\text{Var}_c(E) = \sum_{n=1}^N \left( \frac{n\hbar\omega/2}{\sinh(n\beta\hbar\omega/2)} \right)^2 \simeq \begin{cases} (\hbar\omega)^2 \, e^{-\hbar\omega/(k_B T)} & \text{for } T \ll T_Q \\ \frac{\pi^2}{3} \frac{(k_B T)^3}{\hbar\omega} = \frac{\pi^2}{3} N(k_B T)^2 \left( \frac{T}{T_F} \right) & \text{for } T_Q \ll T \ll T_F \\ N(k_B T)^2 & \text{for } T \gg T_F \end{cases} . \tag{87}$$

In the low temperature regime, the exponential suppression of the fluctuations can be related to the existence of a gap $\hbar\omega$ in the excitation spectrum. The result in the intermediate regime $T_Q \ll T \ll T_F$ has been rewritten in terms of the Fermi temperature to make clear that it corresponds to the classical result $\text{Var}_c(E) \simeq N(k_B T)^2$ (given by the equipartition theorem) multiplied by the small factor $T/T_F \ll 1$. This well-known suppression factor originates from the Pauli principle, which restricts thermal fluctuations to take place in a small window of width $k_B T$ around the Fermi level.

The fluctuations of the potential energy are given by Eq. (66) and (67):

$$\text{Var}_c(E_p) \simeq \begin{cases} \frac{1}{8} N^2 (\hbar\omega)^2 \left[ 1 + 2e^{-2\hbar\omega/(k_B T)} \right] & \text{for } T \ll T_Q \\ \frac{1}{8} N^2 \hbar\omega \, k_B T = \frac{1}{8} N(k_B T)^2 \left( \frac{T_F}{T} \right) & \text{for } T_Q \ll T \ll T_F \\ \frac{1}{2} N(k_B T)^2 & \text{for } T \gg T_F \end{cases} . \tag{88}$$

The finite value $\mathrm{Var}_c(E_p) \sim (N\hbar\omega)^2$ at $T = 0$ is a manifestation of the quantum fluctuations in the ground state, like Eq. (5). The relative classical fluctuations (for $T \gg T_F$) behaves as $\mathrm{Var}_c(E_p)/\left(\overline{E_p}^c\right)^2 \simeq 1/N$, as expected, while the relative quantum fluctuations reach the value $\mathrm{Var}_c(E_p)/\left(\overline{E_p}^c\right)^2 = 2/N^2$ at $T = 0$. The comparison between the potential energy and the total energy is more interesting: in the classical regime ($T \gg T_F$), we have $\mathrm{Var}_c(E_p) \simeq (1/2)\mathrm{Var}_c(E)$, as it should for the harmonic potential. In the regime dominated by quantum correlations ($T \ll T_F$), we have rather $\mathrm{Var}_c(E_p) \gg \mathrm{Var}_c(E)$. This observation has interesting consequences for the correlations of kinetic and potential energies. We express the variance of the total energy $E = E_c + E_p$, and use the fact that $E_c$ and $E_p$ have the same statistical properties for harmonic confinement:

$$\mathrm{Var}_c(E) = 2\,\mathrm{Var}_c(E_p) + 2\,\mathrm{Cov}_c(E_c, E_p)\,. \tag{89}$$

In the classical regime $T \gg T_F$, we have obtained $\mathrm{Var}_c(E_p) = \mathrm{Var}_c(E_c) \simeq (1/2)\mathrm{Var}_c(E)$, which is related to the well-known fact that the kinetic and potential energy are uncorrelated: $\mathrm{Cov}_c(E_c, E_p) \simeq 0$. In the regime $T \ll T_F$, we have obtained that $\mathrm{Var}_c(E_p) \gg \mathrm{Var}_c(E)$, implying that potential and kinetic energies are anti-correlated

$$\frac{\mathrm{Cov}_c(E_c, E_p)}{\sqrt{\mathrm{Var}_c(E_c)\mathrm{Var}_c(E_p)}} \simeq -1 \qquad \text{for } T \ll T_F\,, \tag{90}$$

so that the fluctuations can be related as $\delta E_p \simeq -\delta E_c$.

## 5 Index variance for fermions in a harmonic trap

We now apply the general considerations of sections 2 and 3 to the study of the index $\mathcal{N}_+$, corresponding to the number of fermions on the positive axis. It is given by Eq. (4). This quantity is a linear statistics: it is of the form (10), with $h(x) = \Theta(x)$. Therefore, we can use the results of section 2. In particular, the variance of $\mathcal{N}_+$ is given by Eq. (39), with

$$A_{k,l} = \int_0^\infty \psi_k(x)\psi_l(x)\,\mathrm{d}x\,, \tag{91}$$

and

$$B_k = A_{k,k} = \int_0^\infty \psi_k(x)^2\,\mathrm{d}x = \frac{1}{2}\,. \tag{92}$$

This last relation is exact, due to the symmetry of the potential. Using this result, we can rewrite Eq. (39) as:

$$\mathrm{Var}_{c,g}(\mathcal{N}_+) = \frac{1}{4}\sum_k \overline{n_k}^{c,g} + \frac{1}{4}\left(\sum_k \mathrm{Var}_{c,g}(n_k) + \sum_{k \neq l}\mathrm{Cov}_{c,g}(n_k, n_l)\right) - \sum_{k \neq l}\overline{n_k n_l}^{c,g}(A_{k,l})^2\,. \tag{93}$$

The first term gives the mean number of particles. Moreover the second term has a simple structure thanks to the fact that the matrix elements $B_k$ and $A_{k,k}$ are equal and independent of $k$, in Eq. (92). Note that this property in Eq. (92) is specific to the choice of the observable considered here, namely the index $\mathcal{N}_+$. Writing $N = \sum_k n_k$, the second term can be simply identified as the variance of the total number of particles

$$\mathrm{Var}_{c,g}(N) = \sum_k \mathrm{Var}_{c,g}(n_k) + \sum_{k \neq l}\mathrm{Cov}_{c,g}(n_k, n_l) \tag{94}$$

where obviously $\text{Var}_c(N) = 0$ by definition, while $\text{Var}_g(N)$ is finite. Thus:

$$\text{Var}_{c,g}(\mathcal{N}_+) = \frac{1}{4}\,\overline{N}^{c,g} + \frac{1}{4}\,\text{Var}_{c,g}(N) - \sum_{k \neq l} \overline{n_k n_l}^{c,g}(A_{k,l})^2\,. \tag{95}$$

Before studying into detail the variance of $\mathcal{N}_+$ at any temperature, we will first discuss the limit of high temperature in which the fermions behave as classical particles.

## 5.1 High temperature limit: the Maxwell-Boltzmann regime

We start by considering the simplest limiting case, the limit of high temperature $T \gg T_F$. In this case the fermions can be considered as classical particles as the thermal fluctuations dominate. Therefore, their positions $\{x_n\}$ are independent, and they follow the Maxwell-Boltzmann distribution:

$$\text{Proba}(x_n \in [x, x + \mathrm{d}x]) = \sqrt{\frac{\beta m \omega^2}{2\pi}}\, e^{-\beta m \omega^2 x^2/2}\, \mathrm{d}x = p(x)\,\mathrm{d}x\,. \tag{96}$$

The probability that a particle is in the domain $x > 0$ is $p_+ = \frac{1}{2}$. We now need to distinguish the statistical ensembles:

- In the canonical ensemble, the number $N$ of particles in the trap is fixed. Therefore, the mean number of particles with position $x_n > 0$ is:

$$\overline{\mathcal{N}_+}^c = N p_+ = \frac{N}{2}\,. \tag{97}$$

  To compute the variance, we also need the square of $\mathcal{N}_+$:

$$(\mathcal{N}_+)^2 = \left(\sum_n \Theta(x_n)\right)^2 = \sum_n \Theta(x_n) + \sum_{n \neq m} \Theta(x_n)\Theta(x_m)\,, \tag{98}$$

  from which we deduce:

$$\overline{(\mathcal{N}_+)^2}^c = N p_+ + N(N-1)p_+^2 = \frac{N(N+1)}{4}\,. \tag{99}$$

  From these results, we can deduce the variance:

$$\text{Var}_c(\mathcal{N}_+) = \frac{N}{4}\,. \tag{100}$$

- Let us now consider the grand canonical ensemble in which the number $N$ of fermions fluctuates. In this case, the expressions are simply obtained by averaging Eqs. (97,99) over $N$:

$$\overline{\mathcal{N}_+}^g = \frac{\overline{N}^g}{2}\,, \tag{101}$$

$$\overline{(\mathcal{N}_+)^2}^g = \frac{\overline{N}^g}{4} + \frac{\overline{N^2}^g}{4}\,. \tag{102}$$

From which we deduce:

$$\text{Var}_g(\mathcal{N}_+) = \frac{\overline{N}^g}{4} + \frac{1}{4}\,\text{Var}_g(N)\,, \tag{103}$$

where we have introduced the variance of the total number of particles $\text{Var}_g(N) = \overline{N^2}^g - (\overline{N}^g)^2$. The properties of this variance and its temperature dependence are discussed in Appendix A.

Let us remark that even though the relation between canonical and grand canonical variances (103) has been derived in the classical Maxwell-Boltzmann regime it actually turns out to be much more general, as we discuss in Subsection 5.4 below.

In the limit of high temperature $T \gg T_F$, the index variance thus reads:

$$
\begin{cases}
\text{Var}_c(\mathcal{N}_+) \simeq \dfrac{N}{4} & \text{canonical,} \\[3mm]
\text{Var}_g(\mathcal{N}_+) \simeq \dfrac{\overline{N}^g}{2} & \text{grand canonical.}
\end{cases}
\tag{104}
$$

## 5.2 Canonical ensemble

We have derived in the previous sections a general expression for $\text{Var}(\mathcal{N}_+)$, Eq. (95), which involves the matrix elements $A_{k,l}$ and occupation numbers. The coefficients $A_{k,l}$ are computed in Appendix C, and we studied the occupation numbers in Section 3. In this section we will combine these results to derive a more explicit expression for the index variance of fermions in a harmonic trap $\text{Var}_c(\mathcal{N}_+)$, in the canonical ensemble. In this case, since the total number $N$ of fermions is fixed, $\text{Var}_c(N) = 0$ and Eq. (95) reduces to:

$$
\text{Var}_c(\mathcal{N}_+) = \frac{N}{4} - \sum_{\substack{k,l=0 \\ k \neq l}}^{\infty} \overline{n_k n_l}^c A_{k,l}^2 \, .
\tag{105}
$$

We will compute this variance first in the quantum regime $T \sim T_Q$, then in the thermal regime $T \sim T_F$.

### 5.2.1 Quantum regime

This regime can be reached for finite $\beta = 1/k_B T$ by letting the number $N$ of fermions become large. Since we already know the variance of $\mathcal{N}_+$ at zero temperature, Eq. (5), we will focus on the difference between the variance at temperature $T$ and the one at zero temperature $T = 0$:

$$
\Delta\text{Var}_c(\mathcal{N}_+) = \text{Var}_c(\mathcal{N}_+)|_T - \text{Var}_c(\mathcal{N}_+)|_{T=0} \, .
\tag{106}
$$

At $T = 0$, Eq. (105) is still valid, but with fixed occupation numbers. Only the $N$ lowest energy levels are occupied, thus

$$
n_k = \overline{n_k}^c =
\begin{cases}
1 & \text{if } k < N \, , \\
0 & \text{if } k \geqslant N \, .
\end{cases}
\tag{107}
$$

Therefore, we have from Eq. (105):

$$
\Delta\text{Var}_c(\mathcal{N}_+) = - \sum_{\substack{k,l=0 \\ k \neq l}}^{\infty} \overline{n_k n_l}^c A_{k,l}^2 + \sum_{\substack{k,l=0 \\ k \neq l}}^{N-1} A_{k,l}^2 \, .
\tag{108}
$$

Since the main differences between the occupation numbers at zero and finite temperature are visible near the Fermi level $N-1$, we shift the indices in the sums to start the summation from the Fermi level:

$$
\Delta\text{Var}_c(\mathcal{N}_+) = \sum_{\substack{k,l=0 \\ k \neq l}}^{N-1} (1 - \overline{n_{N-k-1} n_{N-l-1}}^c) A_{N-k-1,N-l-1}^2 - \sum_{\substack{k,l=0 \\ k \neq l}}^{\infty} \overline{n_{N+k} n_{N+l}}^c A_{N+k,N+l}^2
$$
$$
- 2 \sum_{k=0}^{N-1} \sum_{l=0}^{\infty} \overline{n_{N-k-1} n_{N+l}}^c A_{N-k-1,N+l}^2 \, .
\tag{109}
$$

First, for large $N$, we can let the summations go to infinity, as the corrections are exponentially small with $N$. Then, since the coefficients $A_{k,l}$ given by Eq. (168) are non zero only if $k$ and $l$ have different parity, we get:

$$\Delta \mathrm{Var}_c(\mathcal{N}_+) = \frac{1}{\pi^2} \sum_{\substack{k,l=0 \\ \neq \text{ parity}}}^{\infty} (1 - \overline{n_{N-k-1}n_{N-l-1}}^c - \overline{n_{N+k}n_{N+l}}^c) \frac{1}{(k-l)^2}$$

$$- \frac{2}{\pi^2} \sum_{\substack{k,l=0 \\ \text{same parity}}}^{\infty} \overline{n_{N-k-1}n_{N+l}}^c \frac{1}{(k+l+1)^2} \, . \quad (110)$$

We can then use Eq. (53) to evaluate the thermal averages of products of occupation numbers. Introducing $k-l = 2n-1$ in the first sum, $k+l = 2n-2$ in the second one, and making use of Eq. (63) many cancellations occur, yielding a compact expression:

$$\boxed{\Delta \mathrm{Var}_c(\mathcal{N}_+) = F_Q(\beta \hbar \omega) = \frac{2}{\pi^2} \sum_{n=1}^{\infty} \frac{1}{2n-1} \frac{1}{e^{\beta \hbar \omega(2n-1)} - 1}} \quad (111)$$

involving a universal function $F_Q$, as argued below in Section 5.5. This is our final result for the variance in the quantum regime for the canonical ensemble. Remarkably, this formula for fermions involves Bose-Einstein factors, like in the mean total energy [30] or its variance (87). This observation has a simple origin: the system of fermions have particle-hole excitations of bosonic nature. This well-known fact is as the heart of the bosonization technique for 1D Fermi liquids (see [40] for a general reference and [23] for a discussion of bosonization in the presence of a harmonic well).

From our result (111), we can extract the asymptotic behaviours of the temperature dependent part of the variance:

$$\Delta \mathrm{Var}_c(\mathcal{N}_+) \simeq \begin{cases} \dfrac{2}{\pi^2} \, e^{-T_Q/T} & \text{for } T \ll T_Q \\[2ex] \dfrac{T}{4T_Q} & \text{for } T \gg T_Q \, . \end{cases} \quad (112)$$

The low temperature behaviour can be simply associated with the existence of a gap $\hbar \omega$ in the excitation spectrum.

### 5.2.2 Thermal regime

We now consider to the thermal regime, where the temperature is of the order of the Fermi temperature $T \sim T_F$. We fix $y = \beta N \hbar \omega = T_F/T$ and let $N \to \infty$. In this case, we proceed as in Subsection 4.2 where we analysed the potential energy. The study of the index is however more simple thanks to the simplification mentioned in the beginning of Section 5, which makes the second term in parenthesis in Eq. (93) vanish in the canonical ensemble. As a result, in this case it is sufficient to replace the occupation numbers by the rescaled Fermi-Dirac distribution: $\overline{n_k}^c \simeq f_y(k/N) + \mathcal{O}(N^{-1})$ where $f_y$ is given by Eq. (84):

$$\mathrm{Var}_c(\mathcal{N}_+) \simeq \frac{N}{4} - \sum_{k \neq l} f_y \left(\frac{k}{N}\right) f_y \left(\frac{l}{N}\right) (A_{k,l})^2 \, . \quad (113)$$

Let us first rewrite the double sum as:

$$\sum_{k \neq l} f_y \left(\frac{k}{N}\right) f_y \left(\frac{l}{N}\right) (A_{k,l})^2 = 2 \sum_{k=0}^{\infty} \sum_{p=1}^{k} f_y \left(\frac{k}{N}\right) f_y \left(\frac{k-p}{N}\right) (A_{k,k-p})^2 \, . \quad (114)$$

Since $A_{k,l}$ is non zero only if $k$ and $l$ have different parity, see Eq. (168), the sum over $p$ involves only odd integers $p = 2n - 1$. Replacing the summation over $k$ by an integral over $x = k/N$ gives:

$$\sum_{k \neq l} f_y\left(\frac{k}{N}\right) f_y\left(\frac{l}{N}\right) (A_{k,l})^2 \simeq \frac{2N}{\pi^2} \int_0^\infty \sum_{n=1}^\infty \frac{f_y(x)^2}{(2n-1)^2} dx = \frac{N}{4} \int_0^\infty f_y(x)^2 dx \,. \tag{115}$$

Using this result in Eq. (113), along with

$$N = \sum_{k=0}^\infty \overline{n_k}^{\,c} \simeq N \int_0^\infty f_y(x) dx \,, \tag{116}$$

yields

$$\text{Var}_c(\mathcal{N}_+) \simeq \frac{N}{4} \int_0^\infty f_y(x)(1 - f_y(x)) dx = \frac{1}{4} \text{Var}_g(N) \,, \tag{117}$$

where $\text{Var}_g(N)$ is the variance of the total number of particles in the *grand canonical* ensemble, where $\overline{N}^g$ must be replaced by $N$. This non trivial relation between $\text{Var}_c(\mathcal{N}_+)$ and $\text{Var}_g(N)$ relies on the specific properties of the matrix elements $A_{k,l}$, thus on the nature of the observable $\mathcal{N}_+$. Using the expression of this variance from Appendix A, we obtain the final expression for the variance of the index in this regime:

$$\boxed{\text{Var}_c(\mathcal{N}_+) \simeq N \, F_T\left(y = \frac{T_F}{T}\right) = N \, \frac{1 - e^{-y}}{4y}} \tag{118}$$

where the subleading $T = 0$ contribution has been omitted. This is our final result in the thermal regime for the canonical ensemble. From this general expression of the variance, we can extract its asymptotic behaviours as function of the temperature:

$$\text{Var}_c(\mathcal{N}_+) \simeq N \times \begin{cases} \dfrac{T}{4T_F} & \text{for } T \ll T_F \,, \\[2mm] \dfrac{1}{4} & \text{for } T \gg T_F \,, \end{cases} \tag{119}$$

First note that the high temperature limit $T \gg T_F$ matches with the Maxwell-Boltzmann case, Eq. (100), as it should. In addition, the low temperature limit in this thermal regime, $T \ll T_F$, smoothly matches the high temperature limit from the quantum regime, $T \gg T_Q$, Eq. (112). This indicates that there is no intermediate regime of temperature between these two. The low temperature result can be simply understood as the classical result, $N/4$, reduced by the factor $T/T_F$ characteristic of a degenerate Fermi gas [30], as already mentioned for the potential energy.

## 5.3 Grand canonical ensemble

In the previous section we derived expressions for $\text{Var}(\mathcal{N}_+)$ in the canonical ensemble in both quantum and thermal regimes. We now perform a similar computation in the grand canonical ensemble. In this case, the chemical potential $\mu$ is fixed, while the number of fermions in the trap fluctuates. In order to easily compare the results between the two ensembles, we will use the mean number of particles $\overline{N}^g$ as a parameter instead of the chemical potential. The two are related by Eq. (41).

The mean occupation numbers $\overline{n_k}^g$ are given by the Fermi-Dirac distribution (40). Occupations are uncorrelated between different energy levels. This allows to rewrite the general expression (95) as

$$\mathrm{Var}_g(\mathcal{N}_+) = \frac{\overline{N}^g}{4} + \frac{1}{4}\mathrm{Var}_g(N) - \sum_{\substack{k,l=0 \\ k\neq l}}^{\infty} \overline{n_k}^g\overline{n_l}^g A_{k,l}^2 \,, \tag{120}$$

where the variance of the total number of particles $\mathrm{Var}_g(N)$ is studied in Appendix A. As before, we will first discuss the quantum regime and then the thermal one.

### 5.3.1 Quantum regime

Again, we fix $\beta = 1/(k_B T)$ and let $\overline{N}^g \to \infty$. As before, we focus on the difference

$$\Delta\mathrm{Var}_g(\mathcal{N}_+) = \mathrm{Var}_g(\mathcal{N}_+)|_T - \mathrm{Var}_g(\mathcal{N}_+)|_{T=0} \,. \tag{121}$$

Using Eq. (120), we can express this as

$$\Delta\mathrm{Var}_g(\mathcal{N}_+) = \frac{1}{4}\mathrm{Var}_g(N) - \sum_{\substack{k,l=0 \\ k\neq l}}^{\infty} \overline{n_k n_l}^g A_{k,l}^2 + \sum_{\substack{k,l=0 \\ k\neq l}}^{N-1} A_{k,l}^2 \,. \tag{122}$$

We evaluated the same double sums in section 5.2.1, using only relations (11) and (63) which hold in both ensembles. Therefore, our previous derivation is still valid, and we have:

$$-\sum_{\substack{k,l=0 \\ k\neq l}}^{\infty} \overline{n_k n_l}^g A_{k,l}^2 + \sum_{\substack{k,l=0 \\ k\neq l}}^{N-1} A_{k,l}^2 = \Delta\mathrm{Var}_c(\mathcal{N}_+) \,. \tag{123}$$

We obtain the final expression for the variance of the particle number:

$$\boxed{\mathrm{Var}_g(\mathcal{N}_+) = \mathrm{Var}_c(\mathcal{N}_+) + \frac{1}{4}\,\mathrm{Var}_g(N) = \mathrm{Var}(\mathcal{N}_+)|_{T=0} + F_Q(\beta\hbar\omega) + \frac{1}{4}\,\mathrm{Var}_g(N)} \tag{124}$$

where $F_Q(\xi)$ is given by Eq. (111). The variance in the grand canonical ensemble is thus obtained from the canonical one by adding a term proportional to the variance of the total number of particles. Using the limiting behaviours of $\mathrm{Var}_g(N)$ given in Appendix A along with Eq. (112), we can straightforwardly deduce the asymptotic behaviours:

$$\Delta\mathrm{Var}_g(\mathcal{N}_+) \simeq \begin{cases} \dfrac{1}{2}\,\mathrm{e}^{-T_Q/2T} & \text{for } T \ll T_Q\,, \\[2mm] \dfrac{T}{2T_Q} & \text{for } T \gg T_Q\,. \end{cases} \tag{125}$$

We again obtain an essential singularity at zero temperature, but different from the canonical case ($\mathrm{e}^{-T_Q/T}$ *vs* $\mathrm{e}^{-T_Q/2T}$). This is due to the fact that in the grand-canonical case, the leading contribution comes from the term proportional to the total number of particles $\mathrm{Var}_g(N)$.

### 5.3.2 Thermal regime

We now fix $y = \overline{N}^g\beta\hbar\omega$ and let $\overline{N}^g \to \infty$. As in the quantum regime, the last term in Eq. (120) was already computed in section 5.2.2, and is given by Eq. (115). Therefore, we straightforwardly obtain:

$$\boxed{\mathrm{Var}_g(\mathcal{N}_+) = \mathrm{Var}_c(\mathcal{N}_+) + \frac{1}{4}\,\mathrm{Var}_g(N) \simeq \overline{N}^g\,F_T\left(y = \frac{T_F}{T}\right) + \frac{1}{4}\,\mathrm{Var}_g(N)} \tag{126}$$

where $\mathrm{Var}_c(\mathcal{N}_+)$ is here given by Eq. (118) with $N$ substituted by $\overline{N}^g$. In the r.h.s., we have neglected the subleading $T = 0$ contribution, Eq. (5). In this regime, the variance $\mathrm{Var}_g(N)$ can be computed explicitly and is given by Eq. (151):

$$\mathrm{Var}_g(N) \simeq \overline{N}^g \frac{1 - e^{-y}}{y} = 4 \overline{N}^g F_T(y). \tag{127}$$

Therefore, we can rewrite the index variance as

$$\mathrm{Var}_g(\mathcal{N}_+) \simeq 2 \overline{N}^g F_T(y) = 2 \mathrm{Var}_c(\mathcal{N}_+). \tag{128}$$

In this thermal regime, the index variance takes twice its canonical value in the grand canonical case. We can thus straightforwardly obtain its asymptotic behaviours as a function of $T$:

$$\mathrm{Var}_g(\mathcal{N}_+) \simeq \overline{N}^g \times \begin{cases} \dfrac{T}{2T_F} & \text{for } T \ll T_F, \\[2mm] \dfrac{1}{2} & \text{for } T \gg T_F, \end{cases} \tag{129}$$

Again, we check that the low temperature limit $T \ll T_F$ smoothly matches the limit $T \gg T_Q$ in the quantum regime, Eq. (125). We also recover the Maxwell-Boltzmann limit, Eq. (104) in the high temperature limit, as expected, whereas the quantum regime again shows the usual reduction factor $T/T_F$.

## 5.4 Relation between the canonical and the grand canonical variances

We stress here that the relation between the variances in the canonical and grand canonical ensembles is completely general: compare (100) and (103) in the Maxwell-Boltzmann regime, or see Eq. (124) in the quantum regime and Eq. (126) in the thermal regime. This relation can be recovered by a simple heuristic argument: consider an extensive observable of the form $A = \sum_{i=1}^N a_i$. The average is also extensive, and we write it under the form $\overline{A}^g = \overline{N}^g \overline{a}^c$. We write the variance $\mathrm{Var}_g(A) = \mathrm{Var}_g(N a)$ and assume that $a = A/N$ and $N$ are independent. As a result we get:

$$\mathrm{Var}_g(A) = (\overline{N}^g)^2 \mathrm{Var}(a) + (\overline{a}^c)^2 \mathrm{Var}_g(N) = \mathrm{Var}_c(A) + \left(\frac{\partial \overline{A}^c}{\partial N}\right)^2 \mathrm{Var}_g(N). \tag{130}$$

Although this argument is not quite precise, in the case of the energy $E$ of the system, it gives

$$\mathrm{Var}_g(E) = \mathrm{Var}_c(E) + \left(\frac{\partial \overline{E}^c}{\partial N}\right)^2 \mathrm{Var}_g(N), \tag{131}$$

which is a well-known relation in statistical mechanics [29, 30]. Applied to the case of $\mathcal{N}_+$, Eq. (130) becomes:

$$\mathrm{Var}_g(\mathcal{N}_+) = \mathrm{Var}_c(\mathcal{N}_+) + \frac{1}{4} \mathrm{Var}_g(N), \tag{132}$$

which is clearly the relation obtained several times above (103,124,126).

## 5.5 Universality

All our discussion so far has focused on the example of the harmonic trap as in this case the one body wave-functions are known, which makes the computations more explicit. Some of our results can however be extended to any type of confining potential, assuming a single minimum for simplicity.

Let us first discuss the zero temperature result, $\text{Var}(\mathcal{N}_+)|_{T=0}$, given by Eq. (5) for the harmonic trap. A similar result has been obtained for a system of fermions confined in an infinite square well. In this case, the variance of the number of particles in the right half part of the trap can be found in Eq. (40) of Ref. [41]:

$$\text{Var}(\mathcal{N}_+)|_{T=0}^{\text{Box}} \simeq \frac{1}{2\pi^2} \ln N + \frac{1 + \gamma + 2\ln 2}{2\pi^2} \,. \tag{133}$$

The variations of $\mathcal{N}_+$ correspond to particles crossing the origin, thus $\text{Var}(\mathcal{N}_+)|_{T=0}$ measures the quantum fluctuations through the origin. The leading log-terms in (5) and (133) coincide, and thus is a bulk property (which can be related to (30)). However, the subleading constants are different. We thus conclude that they are sensitive to the precise form of the potential. Therefore, we expect only the leading log-term to be universal.

At finite temperature, the fluctuations are controlled by the two scaling functions $F_Q$ and $F_T$, for the quantum and thermal regimes respectively. These functions are determined by the occupation numbers $\overline{n_k}^{\text{c,g}}$, which depend on the spectrum $\{\varepsilon_n\}$ and thus on the potential, and the matrix elements $A_{k,l}$. These latter depend only on the values of the wave functions $\psi_n$ at the center of the trap, see Eq. (165). These can thus be derived by semiclassical methods, such as the WKB approximation, for any kind of potential and exhibit universal properties. In particular, one can show that the expression (168) of these matrix elements for large quantum numbers is universal.

In the quantum regime, the derivation of Sections 5.2.1 and 5.3.1 is based on this universal expression of the matrix elements $A_{k,l}$ and on two properties of the occupation numbers, Eqs. (53) and (63). The first relation is universal, as discussed in Section 3. The second property (63), which implies the symmetry of the occupation numbers around the Fermi level, holds only for a linear spectrum. However, any regular spectrum $\{\varepsilon_n\}$ can be linearised near the Fermi level and thus (63) is also universal in the vicinity of the Fermi level. The temperature scale $T_Q$ is then defined from the gap at the Fermi level:

$$T_Q = \frac{\varepsilon_N - \varepsilon_{N-1}}{k_B} \,. \tag{134}$$

Therefore, the derivation of Sections 5.2.1 and 5.3.1 can be extended to any confining potential, and the scaling function $F_Q$ is thus universal, and given by expression (13).

In the thermal regime, controlled by the scale

$$T_F = \frac{\varepsilon_N + \varepsilon_{N-1}}{2k_B} \,, \tag{135}$$

all the spectrum contributes to the variance $\text{Var}(\mathcal{N}_+)$. Therefore, we do not expect the function $F_T$ to be universal. This can be shown explicitly from the relation (117) which states that this scaling function is proportional to the variance of the total particle number in the grand canonical ensemble: $F_T(T_F/T) = \text{Var}_g(N)/4$. We do not know this function explicitly, however we show in Appendix A that its limiting behaviours are only controlled by the exponent governing the one body density of states $\rho(\varepsilon) \propto \varepsilon^{\alpha-1}$:

$$F_T\left(y = \frac{T_F}{T}\right) \simeq \frac{1}{4} \begin{cases} 1 & \text{for } T \gg T_F \,, \\ \alpha\dfrac{T}{T_F} & \text{for } T \ll T_F \,, \end{cases} \tag{136}$$

which thus depends explicitly on $\alpha$, i.e. is not universal.

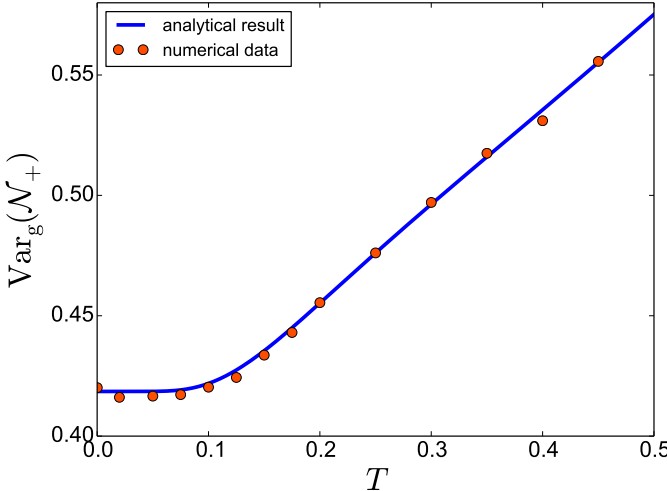

Figure 3: Variance of the index number $\mathcal{N}_+$, in the grand canonical ensemble for $\overline{N}^g = 100$ as a function of temperature (we set $\hbar = \omega = m = k_B = 1$). The points are obtained by averaging over $10^5$ realisations, while the solid line corresponds to our analytical result, Eq. (138).

## 5.6  Numerical simulations

We now compare our results with numerical simulations. Generating numerically realisations of the positions of the fermions is quite difficult. But in the grand canonical ensemble, we can use the fact that the positions of the fermions form a determinantal point process [11,33,42], with kernel:

$$K_T(x, x') = \sum_k \overline{n_k}^g \psi_k(x) \psi_k(x'),\qquad(137)$$

where $\psi_k$ are the one-particle wave functions of the harmonic oscillator (64). We reproduce in Appendix D an algorithm described in Refs. [43,44] which allows to sample such point processes. Therefore, we performed simulations only in the grand canonical case.

We generate realisations of the positions of fermions and compute the corresponding value of $\mathcal{N}_+$ for each of them. From the set of data, we compute numerically $\mathrm{Var}(\mathcal{N}_+)$. We studied the quantum regime, with $\overline{N}^g = 100$. We did not investigate the thermal regime because the computational efforts increase with the temperature, as more energy levels contribute. This makes it difficult to obtain enough statistics in the thermal regime where $T = \mathcal{O}(\overline{N}^g)$. Our result in the grand canonical ensemble (124), combined with the previously known zero temperature expression (5) read:

$$\mathrm{Var}_g(\mathcal{N}_+) \simeq \frac{1}{4}\,\mathrm{Var}_g(N) + F_Q(\beta\hbar\omega) + \frac{1}{2\pi^2}\ln\overline{N}^g + c\,,\qquad(138)$$

where $F_Q(\beta\hbar\omega)$ is given by Eq. (111) and $c$ is a constant, see Eq. (5). We used this expression to compare the numerical data to our analytical result. They show an excellent agreement, see Fig. 3.

## 6  Conclusion

In this paper, we have introduced a general method to study certain many body observables of the form of sums of one body observables of the positions of fermions in a confining trap. We have applied our method to the study of the number $\mathcal{N}_+$ of fermions on the positive axis,

in the case of a harmonic well. We have obtained explicit expressions for the variance of these observables, both in the quantum regime $T \sim T_Q = \hbar\omega/k_B$ and the thermal regime $T \sim T_F = N\hbar\omega/k_B$ in terms of two scaling functions, one universal and the other not. We have shown that these expressions smoothly match. We note that the fluctuations of linear statistics were recently studied by Johansson and Lambert in the grand canonical ensemble [45]. They considered the case where each of the fermions positions $x_i$'s are scaled like $N^{-\delta}$ while the temperature scales like $T \sim N^\alpha$. The results presented in this paper thus corresponds to $\delta = 0$ (which in their terminology corresponds to "macroscopic scale"). In addition, here, we have studied the crossover between the quantum regime ($\alpha = 0$ in the notation of Ref. [45]) and the thermal regime ($\alpha = 1$).

We have emphasised the difference between the statistical ensemble by computing the variance of $\mathcal{N}_+$ both in the canonical and grand-canonical ensembles. This difference is due to the fact that we have considered the fluctuations of a *global* observable, $\mathcal{N}_+ = \sum_n \Theta(x_n)$. On the other hand, *local* quantities, such as the two point density-density correlation functions, are ensemble independent [11].

The computation of these fluctuations in the microcanonical ensemble is still an open question. Indeed, our derivation made extensive use of relation (11) which no longer holds in the microcanonical case. We have studied the first moments of the distribution of the observable $\mathcal{N}_+$ at finite temperature. Computing the full distribution of $\mathcal{N}_+$ would be an interesting but much more challenging question.

## Acknowledgements

This research was supported by ANR grant ANR-17-CE30-0027-01 RaMaTraF. We acknowledge stimulating discussions with Olivier Giraud. *Note added in Proofs:* We thank Kurt Schönhammer for bringing to our attention the recent Ref. [50] where relation (11, 53) was derived in the fermionic case. This paper also reports Eq. 63, first obtained in [51].

## A  Variance of the total particle number in the grand canonical ensemble for any confining potential

As discussed in Section 3, in the grand-canonical ensemble the mean occupation numbers $\overline{n_k}^g$ are given by the Fermi-Dirac distribution, Eq. (40). In addition, occupations are uncorrelated: $\overline{n_k n_l}^g = \overline{n_k}^g \overline{n_l}^g$. This allows to easily evaluate the first moments of the fluctuating total number of particles $N = \sum_k n_k$:

$$\overline{N}^g = \sum_k \overline{n_k}^g = \sum_k \frac{1}{e^{\beta(\varepsilon_k - \mu)} + 1} \, . \tag{139}$$

This first relation links the mean number of particles $\overline{N}^g$ to the chemical potential $\mu$. The variance reads:

$$\mathrm{Var}_g(N) = \sum_k \mathrm{Var}_g(n_k) + \underbrace{\sum_{k \neq l} \mathrm{Cov}_g(n_k, n_l)}_{=0} = \sum_k \overline{n_k}^g (1 - \overline{n_k}^g) \, . \tag{140}$$

Therefore, we obtain:

$$\mathrm{Var}_g(N) = \sum_k \frac{1}{e^{\beta(\varepsilon_k - \mu)} + 1} \left( 1 - \frac{1}{e^{\beta(\varepsilon_k - \mu)} + 1} \right) \, . \tag{141}$$

We can use these expressions to study the limiting behaviours of this variance for any confining potential of the form $V(x) \propto |x|^p$. The spectrum can be determined from a WKB approximation

$$\int_{-x_t}^{x_t} \sqrt{2m(\varepsilon_n - V(x))} = \left(n + \frac{1}{2}\right)\pi\hbar \,, \tag{142}$$

where $x_t$ is the turning point $V(x_t) = \varepsilon_n$. This condition gives the scaling

$$\varepsilon_n \sim n^{1/\alpha}, \quad \alpha = \frac{1}{2} + \frac{1}{p} \,. \tag{143}$$

We can check that $\alpha = 1$ for the harmonic potential while $\alpha = 1/2$ corresponds to the infinite square well. In the continuum limit, this spectrum gives a density of states of the form

$$\rho(\varepsilon) = A\frac{\varepsilon^{\alpha-1}}{\delta^\alpha} \,, \tag{144}$$

where $\delta$ is an energy scale and $A$ is a dimensionless parameter. For a trap containing on average $\overline{N}^g$ particles, the Fermi energy is given by $\varepsilon_F = (\alpha\overline{N}^g/A)^{1/\alpha}\delta$. Therefore, we define the Fermi temperature as

$$T_F = \frac{\varepsilon_F}{k_B} = \left(\frac{\alpha\overline{N}^g}{A}\right)^{1/\alpha}\frac{\delta}{k_B} \,. \tag{145}$$

The temperature scale $T_Q$ can be defined from the gap at the Fermi level (134), which gives

$$T_Q = \frac{1}{k_B\rho(\varepsilon_F)} = \frac{1}{\alpha\overline{N}^g}\,T_F \,. \tag{146}$$

In the quantum regime, the spectrum can be linearised near the Fermi level:

$$\varepsilon_{\overline{N}^g+n} \simeq \varepsilon_F + n\,k_B T_Q \,. \tag{147}$$

The variance $\mathrm{Var}_g(N)$ is thus universal in this regime. In the low temperature limit $T \ll T_Q$, Eq. (139) imposes that the chemical potential is fixed to the middle of the gap $\mu \simeq \varepsilon_F + k_B T_Q/2$. Using this value in Eq. (141), we can study the low temperature limit of the variance of $N$. The leading contribution comes from the two levels $\varepsilon_{\overline{N}^g-1}$ and $\varepsilon_{\overline{N}^g}$ which are the closest to the chemical potential $\mu$. This gives

$$\mathrm{Var}_g(N) \simeq 2\,\mathrm{e}^{-T_Q/2T} \,, \quad T \ll T_Q \,. \tag{148}$$

In the thermal regime $T \sim T_F$, the sums can be replaced by integrals over the energy $\varepsilon$. Eq. (139) becomes:

$$\overline{N}^g = \sum_k \overline{n_k}^g \simeq \int_0^\infty \frac{\rho(\varepsilon)\,\mathrm{d}\varepsilon}{\mathrm{e}^{\beta(\varepsilon-\mu)}+1} = -\frac{A}{(\beta\delta)^\alpha}\Gamma(\alpha)\,\mathrm{Li}_\alpha(-\mathrm{e}^{\beta\mu}) \,, \tag{149}$$

where we recall the function $\mathrm{Li}_\alpha(z) = \sum_{k=1}^\infty z^k/k^\alpha$. This last relation is more conveniently expressed in terms of dimensionless variables:

$$-y^{-\alpha}\,\Gamma(\alpha+1)\,\mathrm{Li}_\alpha(-\mathrm{e}^{y\tilde{\mu}}) = 1 \,, \quad y = \frac{T_F}{T} \text{ and } \tilde{\mu} = \frac{\mu}{\varepsilon_F} \,. \tag{150}$$

This relation fixes the rescaled chemical potential $\tilde{\mu}$ in terms of the rescaled inverse temperature $y$. A similar computation for the variance (141) gives:

$$\mathrm{Var}_g(N) = \frac{1}{\beta}\frac{\partial\overline{N}^g}{\partial\mu} = -\overline{N}^g\,y^{-\alpha}\,\Gamma(\alpha+1)\mathrm{Li}_{\alpha-1}(-\mathrm{e}^{y\tilde{\mu}}) \,. \tag{151}$$

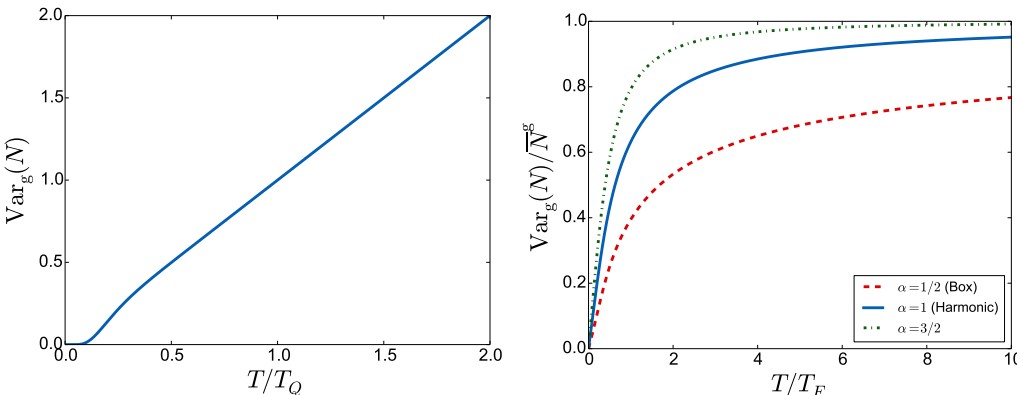

Figure 4: Variance of the total number $N$ of fermions in the grand canonical ensemble, as a function of the temperature $T$. Left: quantum regime $T \sim T_Q$ obtained from (141). This function is universal. Right: thermal regime $T \sim T_F$, given by Eq. (151) for different types of confinement: $\alpha = 1$ (harmonic), $\alpha = 1/2$ (infinite well) and $\alpha = 3/2$.

These two relations (150) and (151) allow to plot this variance for different confining potentials, corresponding to different values of $\alpha$, see Fig. 4, right. In the high-temperature limit, the variance reaches the classical limit

$$\mathrm{Var}_g(N) \simeq \overline{N}^g, \quad T \gg T_F. \tag{152}$$

The other limiting case is:

$$\mathrm{Var}_g(N) \simeq \alpha \overline{N}^g \frac{T}{T_F} = \frac{T}{T_Q}, \quad T_Q \ll T \ll T_F, \tag{153}$$

which shows once again the reduction factor $T/T_F$, as usual in the degenerate Fermi gas [30]. See Fig. 4 for plots of this variance $\mathrm{Var}_g(N)$ in both the quantum and thermal regimes, for different confining potentials. In the case of the harmonic trap ($\alpha = 1$), Eqs. (150,151) reduce to the simple expression

$$\mathrm{Var}_g(N) = \frac{1 - \mathrm{e}^{-y}}{y}, \quad y = \frac{T_F}{T}. \tag{154}$$

## B  Saddle point estimate

Let us consider integrals of the form:

$$I(N) = \int_{\mathscr{C}} \mathrm{d}z \, g(z) \, \mathrm{e}^{-N\phi(z)}, \tag{155}$$

where $\mathscr{C}$ is any contour in the complex plane, $g$ and $\phi$ are any given smooth functions. We want to estimate this integral in the limit of large $N$ by using a saddle point method. The saddle point $z_\star$ is given by

$$\phi'(z_\star) = 0. \tag{156}$$

Let us make the change of variable $z = z_\star + t/\sqrt{N}$ and deform the contour $\mathscr{C}$ such that $t$ is real. We can expand $\phi$ and $g$ near $z_\star$:

$$\phi(z) = \phi(z_\star) + \frac{t^2}{2N}\phi''(z_\star) + \frac{t^3}{6N^{3/2}}\phi^{(3)}(z_\star) + \frac{t^4}{24N^2}\phi^{(4)} + \mathcal{O}(N^{-5/2}), \tag{157}$$

$$g(z) = g(z_\star) + \frac{t}{\sqrt{N}}g'(z_\star) + \frac{t^2}{2N}g''(z_\star) + \mathcal{O}(N^{-3/2}). \tag{158}$$

Using these expansions, the integral $I(N)$ becomes:

$$\begin{aligned}
I(N) = \frac{\mathrm{e}^{-N\phi(z_\star)}}{\sqrt{N}} \int_{\mathbb{R}} \mathrm{d}t \, & \mathrm{e}^{-t^2\phi''(z_\star)/2} \\
& \times \left(1 - \frac{t^3}{6\sqrt{N}}\phi^{(3)}(z_\star) - \frac{t^4}{24N}\phi^{(4)}(z_\star) + \frac{t^6}{72N}(\phi^{(3)}(z_\star))^2 + \mathcal{O}(N^{-3/2})\right) \\
& \times \left(g(z_\star) + \frac{t}{\sqrt{N}}g'(z_\star) + \frac{t^2}{2N}g''(z_\star) + \mathcal{O}(N^{-3/2})\right).
\end{aligned} \tag{159}$$

Computing the Gaussian integrals yields:

$$\begin{aligned}
I(N) = \mathrm{e}^{-N\phi(z_\star)}\sqrt{\frac{2\pi}{N\phi^{(2)}(z_\star)}}\Bigg[ & g(z_\star) + \frac{1}{N}\left(\frac{g^{(2)}(z_\star)}{2\phi^{(2)}(z_\star)} - \frac{g(z_\star)\phi^{(4)}(z_\star)}{8(\phi^{(2)}(z_\star))^2}\right. \\
& \left. + \frac{5g(z_\star)(\phi^{(3)}(z_\star))^2}{24(\phi^{(2)}(z_\star))^3} - \frac{g'(z_\star)\phi^{(3)}(z_\star)}{2(\phi^{(2)}(z_\star))^2}\right) + \mathcal{O}(N^{-2})\Bigg].
\end{aligned} \tag{160}$$

We used this expression in section 3.2.2 to obtain the $\mathcal{O}(N^{-1})$ corrections to the covariance $\overline{n_k n_l}^c - \overline{n_k}^c\overline{n_l}^c$.

## C  Matrix elements

In this section we compute the coefficients $A_{k,l}$ associated to the index variance, Eq. (91), which are obtained from the quantum average. Since the index $\mathcal{N}_+$ is a linear statistics (10) for $h(x) = h(x)^2 = \Theta(x)$, we can straightforwardly obtain the "diagonal terms"

$$B_k = A_{k,k} = \int_0^\infty \psi_k(x)^2 \mathrm{d}x = \frac{1}{2}. \tag{161}$$

The "off-diagonal" coefficients are given by:

$$A_{k,l} = \int_0^\infty \psi_k(x)\psi_l(x)\mathrm{d}x. \tag{162}$$

This integral can be computed analytically. Indeed, let us compute the derivative of $\psi_k(x)\psi'_l(x) - \psi'_k(x)\psi_l(x)$:

$$\frac{\mathrm{d}}{\mathrm{d}x}\left(\psi_k(x)\psi'_l(x) - \psi'_k(x)\psi_l(x)\right) = \psi_k(x)\psi''_l(x) - \psi''_k(x)\psi_l(x) \tag{163}$$

since the other terms cancel out. Using now that $\psi_k$ is solution of the Schrödinger equation $-\frac{\hbar^2}{2m}\psi''_k + V\psi_k = \varepsilon_k\psi_k$, we get:

$$\frac{\mathrm{d}}{\mathrm{d}x}\left(\psi_k(x)\psi'_l(x) - \psi'_k(x)\psi_l(x)\right) = \frac{2m}{\hbar^2}(\varepsilon_k - \varepsilon_l)\psi_k(x)\psi_l(x). \tag{164}$$

This relation allows us to directly compute the integral

$$A_{k,l} = \int_0^\infty \psi_k(x)\psi_l(x)\,\mathrm{d}x = -\frac{\hbar^2}{m}\frac{1}{\varepsilon_k - \varepsilon_l}\left(\psi_k(0)\psi_l'(0) - \psi_k'(0)\psi_l(0)\right). \tag{165}$$

Using the expression of the wave functions, Eq. (64) and properties of the Hermite polynomials [46], we obtain that $A_{k,l}$ is zero if $k$ and $l$ have the same parity, and

$$A_{2m,2n+1} = \frac{(-2)^{n+m+1}\Gamma(m+\frac{1}{2})\Gamma(n+\frac{3}{2})}{\pi\sqrt{2\pi}\sqrt{(2n+1)!(2m)!}(2m-2n-1)}. \tag{166}$$

Since we only need the square of these coefficients, we can simplify this expression using properties of the $\Gamma$ function:

$$(A_{2m,2n+1})^2 = \frac{\Gamma(m+\frac{1}{2})\Gamma(n+\frac{3}{2})}{\pi^2 n! m!}\frac{1}{(2m-2n-1)^2}. \tag{167}$$

We are interested in the limit in which the number $N$ of fermions is large. We expect the variance of $\mathcal{N}_+$ to be dominated by the fluctuations near the Fermi level. Therefore, it is enough to estimate the coefficients $A_{k,l}$ for $k$ and $l$ of order $N$. Therefore, for $k = Nt$ and $l = k + p$, we get:

$$(A_{Nt,Nt+p})^2 \simeq \begin{cases} 0 & \text{if } p \text{ is even,} \\ \dfrac{1}{p^2\pi^2} & \text{if } p \text{ is odd.} \end{cases} \tag{168}$$

Note that these are the leading contributions to the coefficients $A_{k,l}$: they also receive $\mathcal{O}(N^{-1})$ corrections. In addition, we have only considered the case where $k,l = \mathcal{O}(N)$ with $k-l = \mathcal{O}(1)$. It is clear from the final expression of $A_{k,l}$ (168) that the case $k-l = \mathcal{O}(N)$ gives only subleading corrections.

## D  Numerical simulations of determinantal point processes

A determinantal point process is a random point process $\{x_n\}$ which is entirely characterised by a *kernel* $K(x,y)$. All $n$-points correlations functions can be expressed as $n \times n$ determinants involving the kernel $K$, see Eq. (25). We consider such a process with a kernel

$$K(x,y) = \sum_{k=0}^\infty \lambda_k \psi_k^*(x)\psi_k(y), \tag{169}$$

where $0 \leqslant \lambda_k \leqslant 1$ and

$$\int_{-\infty}^\infty \psi_k^*(x)\psi_l(x)\mathrm{d}x = \delta_{k,l}. \tag{170}$$

A general method to generate numerically realisations of this process was introduced in Ref. [43]. We reproduce here a similar algorithm described in [44]. This algorithm was also used in the physics literature, see e.g. Refs. [47, 48]. It relies on the following theorem: introduce a set of Bernoulli random variables $n_k = 0$ or $1$, with mean value $\overline{n_k} = \lambda_k$. Then, the determinantal point process with kernel

$$\tilde{K}(x,y) = \sum_{k=0}^\infty n_k \psi_k^*(x)\psi_k(y) \tag{171}$$

has the same statistics as the original process with kernel (169). In terms of fermions, this means that picking a realisation of the positions of the particles in the grand canonical ensemble is equivalent to first picking a quantum state $\{n_k\}$ from the Gibbs measure (6) and then generating a realisation of the positions from that quantum state. Using this property, one can generate realisations of the determinantal point process using the following procedure:

1. Generate the index $M$ of the highest occupied level, using that

$$\text{Proba}(M = m) = \lambda_m \prod_{i > m}(1 - \lambda_i). \tag{172}$$

2. Generate the occupation numbers for $k < M$, from the measure

$$\text{Proba}(n_k = 1) = \lambda_k. \tag{173}$$

Set $n_M = 1$ and $n_p = 0$ for $p > M$. Note that this realisation will contain $N = \sum_k n_k$ points. Denote also $\{k_n\}_{n=1,\dots,N}$ the indices of the occupied levels ($n_{k_i} = 1$) and $\vec{v}(x) = (\psi_{k_1}(x), \dots, \psi_{k_N}(x))^T$.

3. Pick the first point $X_1$ from the distribution

$$p_1(x) = \frac{1}{N} \sum_{p=1}^{N} \left| \psi_{k_p}(x) \right|^2 = \frac{||\vec{v}(x)||^2}{N}, \tag{174}$$

and introduce $\vec{e}_1 = \vec{v}(X_1)/||\vec{v}(X_1)||$.

4. Knowing the positions $\{X_1, \dots, X_n\}$ of the first $n$ points and the set of unit vectors $(\vec{e}_1, \dots, \vec{e}_n)$ generate the position $X_{n+1}$ of the next point from the distribution

$$p_{n+1}(x) = \frac{1}{N-n} \left( ||\vec{v}(x)||^2 - \sum_{j=1}^{n} \left| \vec{e}_j^* \cdot \vec{v}(x) \right|^2 \right). \tag{175}$$

Construct $\vec{e}_{n+1} = \vec{w}_{n+1}/||\vec{w}_{n+1}||$, where

$$w_{n+1} = \vec{v}(X_{n+1}) - \sum_{j=1}^{n} (\vec{e}_j^* \cdot \vec{v}(X_{n+1})) \, \vec{e}_j. \tag{176}$$

This procedure gives a realisation $(X_1, \dots, X_N)$ of the determinantal point process with kernel (169). Generating the points from the rather complex distributions $p_n(x)$ can be done using rejection sampling [49].

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
