# Peer review of "Fluctuations of observables for free fermions in a harmonic trap at finite temperature"

_SciPost Physics, doi:SciPost Phys. 4, 014 (2018)_

## Round 1 · Referee Report · Anonymous · 2018-1-8

Strengths

1. simple problem with physically interesting question
2. clear presentation

Weaknesses

1. results could have been further discussed

Report

The authors study fluctuations properties of a free Fermi gas
trapped in a harmonic potential at finite temperature.
More precisely, they consider the so-called linear statistics,
i.e. the sum of an arbitrary function of the fermion coordinates.
The main result of the manuscript is an explicit formula for
the variance of the linear statistics, Eq. (38), which is valid
for both canonical and grand canonical ensembles.
This result is then applied to calculate the fluctuations
of the particle number in the right half of the trap.
The scaling functions in the two different, quantum and thermal
regimes are explicitly calculated. In the quantum regime the
analytical result is compared against numerical simulations
with a good agreement.

I believe that, despite the simple, textbook-like calculations,
the manuscript deals with a physically interesting problem.
In particular, the finite temperature results on the particle
fluctuation are new, and the general result on linear statistics
possibly has a broader range of applicability.
Therefore I recommend the publication of the manuscript after the
authors have considered the issue raised below.

Requested changes

What I was missing a bit is a discussion about the role of the
potential. In fact, one could have equally well asked about the
fluctuations in one half of a finite box with no external potential.
At zero temperature this question has already been studied, see
EPL 98, 20003 (2012) and Phys. Rev. B 82, 012405 (2010).
Remarkably, it turns out that the box-result at $T=0$ is
identical to Eq. (5) of the manuscript, i.e. the role of the
potential is irrelevant. This naturally raises the question,
what is the situation at finite temperature? Since the precise
form of the eigenfunctions enters only through integrals in
Eqs. (90-91), one could expect some universality also at finite T.
It would be nice, if the authors could comment about this.

  • validity: high
  • significance: good
  • originality: good
  • clarity: high
  • formatting: excellent
  • grammar: perfect

Author:  Aurélien Grabsch  on 2018-02-09  [id 210]

(in reply to Report 1 on 2018-01-08)
Category:
answer to question

We are thankful to the referee for his/her careful reading of the manuscript and his/her positive comments.

The referee has raised an interesting question about the universality of our results, which were derived for a harmonic confinement.

I) Concerning the zero temperature result: although the referee is right about the universality of the dominant logarithmic term of the variance, we stress that the subleading constant is potential dependent. For a hard box confinement, the zero temperature result for $\mathcal{N}_+$ can be found for example in Eq. (40) of EPL 98, 20003 (2012) and reads

\[ \mathrm{Var}(\mathcal{N}_+) |_{T=0}^\text{Box} = \frac{1}{2\pi^2} \ln N + \frac{1+\gamma+2\ln 2}{2\pi^2} \:, \]

which differs from Eq. (5) by a constant term $\ln 2/(2\pi^2)$.

II) Nevertheless, at finite temperature, stimulated by the question of the referee we have investigated further the universality of our formulae. We have now added a new section (5.5) and extended the discussion of Appendix A. The main outcome is: - In the quantum regime the thermal fluctuations involve excitations around the Fermi level. For a one dimensional smooth confining potential, the spectrum is regular and can be linearlised near the Fermi level. The fluctuations are thus described by the same function $F_Q$ as for the harmonic oscillator. - In the thermal regime all the spectrum contributes and therefore the results are not universal.

---

## Round 1 · Referee Report · Anonymous · 2018-1-9

Strengths

1) Very neat presentation of original results

2) Calculations and logic easy to follow

Weaknesses

1) Focus only on the harmonic potential

Report

The authors develop a general method to calculate the variance of linear statistics in a free one-dimensional Fermi gas, with a confining potential (Eq.(38) in particular). They apply the formalism to determine the variance of the number of fermions in the positive semi-line (index) both in the canonical and grand canonical ensembles. The analysis is limited to the harmonic potential.

The authors also recover the variance of the potential energy previously derived in Ref. [27]. They conclude with a numerical simulation of the determinantal process associated to the fermion positions in an harmonic potential that further confirms their findings in the grand canonical ensemble.

I found the paper pedagogically written and interesting. Since moreover to my best knowledge the results presented in Sec.2.2, 3.2.1 and 4-5 are original and correct, I recommend the paper to be published after the authors will address the minor points below.

Requested changes

1)The formalism allows calculating in particular the variance of N_+ in the Ground State for an arbitrary potential. Does this show a Log(N) term as in Eq.(5)? If yes, is this term universal (i.e. potential independent)?

2)At pag. 13: There is a typo in the penultimate sentence of sec. 3.1

3)At pag. 22: kinetic->potential, (although they are equivalent seems more consistent with the text to continue discussing potential energy)

4)At pag. 22: formula (113) an "f_y" is missing after the second sum

5)At pag. 23: Formulas (119)-(120). I think "c" has to be replaced with "g", saying perhaps that what is calculated in (120) is the variance in the canonical ensemble with N replaced by \bar{N}_g. Correct?

6)It seems that the Bose factor that appears in Eq.(110) is independent from the choice of the potential. Perhaps the authors can stress this fact and its implications for the asymptotics (111) in the quantum regime. Are then these asymptotics expected to be potential independent?

  • validity: top
  • significance: good
  • originality: good
  • clarity: top
  • formatting: perfect
  • grammar: excellent

Author:  Aurélien Grabsch  on 2018-02-09  [id 211]

(in reply to Report 2 on 2018-01-09)
Category:
answer to question

We are grateful to the referee for his/her very careful reading of the manuscript and his/her positive report.

1) and 6) The referee has raised the same interesting points as Referee 1. Here is our answer: I) Concerning the zero temperature result: although the referee is right about the universality of the dominant logarithmic term of the variance, we stress that the subleading constant is potential dependent. For a hard box confinement, the zero temperature result for $\mathcal{N}_+$ can be found for example in Eq. (40) of EPL 98, 20003 (2012) and reads

\[ \mathrm{Var}(\mathcal{N}_+) |_{T=0}^\text{Box} = \frac{1}{2\pi^2} \ln N + \frac{1+\gamma+2\ln 2}{2\pi^2} \:, \]

which differs from Eq. (5) by a constant term $\ln 2/(2\pi^2)$.

II) Nevertheless, at finite temperature, stimulated by the question of the referee we have investigated further the universality of our formulae. We have now added a new section (5.5) and extended the discussion of Appendix A. The main outcome is: - In the quantum regime the thermal fluctuations involve excitations around the Fermi level. For a one dimensional smooth confining potential, the spectrum is regular and can be linearlised near the Fermi level. The fluctuations are thus described by the same function $F_Q$ as for the harmonic oscillator. - In the thermal regime all the spectrum contributes and therefore the results are not universal.

2-5) We thank the referee for pointing out these typos.

---

## Round 2 · Author Response

Please find enclosed the revised version of our paper

"Fluctuations of observables for free fermions in a harmonic trap at finite temperature"

Our manuscript was reviewed by two referees.

Both of them wrote very positive reports and suggested the publication of our paper in SciPost. Their main concern was about the
universality of our results, beyond the case of the harmonic trap which is exactly solved in the present manuscript. We have added a new
section (5.5) and extended the discussion of Appendix A to address the question of universality.

We hope that you will receive positively our resubmission.

---

## Round 2 · List of Changes

• We have added a section (5.5) to discuss the universality of our results with respect to the choice of the potential;

  • We have generalised the discussion of Appendix A about the variance of the total number of particles in the grand canonical ensemble to other types of potentials (not only harmonic);

  • We have corrected the typos pointed out by referee 2;

  • We have slightly changed the title and abstract to make them more intelligible for non specialists (we replaced "linear statistics" by "observables").

---

## Editorial Decision

published